# Subcellular connectomic analyses of energy networks in striated muscle

Christopher K.E. Bleck[1], Yuho Kim[1], T. Bradley Willingham[1] & Brian Glancy [1,2]

Mapping biological circuit connectivity has revolutionized our understanding of structure-function relationships. Although connectomic analyses have primarily focused on neural systems, electrical connectivity within muscle mitochondrial networks was recently demonstrated to provide a rapid mechanism for cellular energy distribution. However, tools to evaluate organelle connectivity with high spatial fidelity within single cells are currently lacking. Here, we developed a framework to quantitatively assess mitochondrial network connectivity and interactions with cellular sites of energy storage, utilization, and calcium cycling in cardiac, oxidative, and glycolytic muscle. We demonstrate that mitochondrial network configuration, individual mitochondrial size and shape, and the junctions connecting mitochondria within each network are consistent with the differing contraction demands of each muscle type. Moreover, mitochondria-lipid droplet interaction analyses suggest that individual mitochondria within networks may play specialized roles regarding energy distribution and calcium cycling within the cell and reveal the power of connectomic analyses of organelle interactions within single cells.

---

[1] National Heart, Lung, and Blood Institute, National Institutes of Health, Bethesda, MD 20892, United States. [2] National Institute of Arthritis and Musculoskeletal and Skin Diseases, National Institutes of Health, Bethesda, MD 20892, United States. Correspondence and requests for materials should be addressed to B.G. (email: brian.glancy@nih.gov)

Connectivity among cellular structures is of great interest to biologists as these physical interactions facilitate the coordinated movement of ions, molecules, and/or proteins which provides the basis for many cellular processes. The field of connectomics has grown in the last decade due in large part to advances in three-dimensional (3D) electron microscopy[1,2] which has allowed for high spatial resolution analyses of structure–function relationships. However, connectomic studies have thus far been primarily limited to neural networks[3] despite the well-known importance of cell–cell or organelle–organelle interactions for cell signaling, calcium cycling, protein trafficking, and other cellular processes in many cell types[4–6]. While the use of 3D electron microscopy in non-neural tissues has been increasing over the past few years[7–14], its utility has been severely limited by the lack of straightforward tools available to quickly assess specific subcellular structures within these large 3D datasets. As a result, network scale analyses of connections and interactions between subcellular organelles have thus far relied on light microscopic techniques on cultured or isolated cells[15,16] which often lack the spatial resolution to accurately assess organelle–organelle contacts of 30 nm or less in size[17–19].

To overcome these limitations, we utilized a connectomics approach (Supplementary Figure 1) to interrogate intra- and inter-organelle interactions on the cellular network scale and focused on the connectivity of muscle mitochondrial networks which, as we recently showed, provides a mechanism for rapid cellular energy distribution[20]. We find that muscle mitochondrial networks can take parallel, perpendicular, or grid-like configurations depending on muscle fiber type and that the orientation, size, and degree of structural and electrical connectivity of the muscle mitochondrial reticulum are consistent with the functional demands of the cell. High throughput analyses of individual mitochondria within each network and of the intermitochondrial junctions structurally connecting the network reveal that the individual components within mitochondrial networks are also consistent with the energetic and calcium cycling demands of striated muscle cells. Finally, by evaluating the interactions between mitochondria and other cellular components, we find that mitochondria connected to lipid droplets may represent a specialized pool of mitochondria with a greater capacity for energy distribution while non-lipid droplet connected mitochondria may be specialized for calcium cycling. Thus, subcellular connectomics provides a powerful platform for assessing the organelle interactions critical to cellular function.

## Results

### Development of connectomics image segmentation approach.

After optimizing the in vivo tissue fixation and staining procedures to enhance the contrast of mitochondria and other membranous organelles (Supplementary Figure 1a), 3D tissue structures were collected from striated mouse muscles across a ~10-fold range of mitochondrial contents by either focused ion beam scanning electron microscopy (FIB-SEM, 5–10 nm isotropic voxels, Supplementary Movie 1) or serial block face scanning electron microscopy (SBF–SEM, $6 \times 6 \times 35$ nm anisotropic voxels). In order to quantitatively assess mitochondrial and other cellular structures, accurate segmentation of each structure within the EM volumes was required. This step is a major limitation to the application of large scale 3D electron microscopy to assess organelle connectivity and interactions on the network scale as the primary method currently in use is to trace cellular structures by hand[7–14]. While manual segmentation is very accurate, it is also extremely time consuming as segmentation can take hundreds to thousands of hours[21,22] depending on the size of the 3D

datasets and the number and type of cellular structures to be assessed. Segmentation based on image intensity thresholding combined with spatial filtering is relatively fast and can be sufficient to provide a rough, qualitative overview of cellular structures in some cases[20], however, the accuracy of this method is insufficient for quantitative analysis of organelle morphology or connectivity. To achieve high throughput segmentation of cellular structures in large 3D volumes, we utilized a straightforward automated segmentation approach to separate all visually discernable components within the muscle cells. After pre-processing the 3D datasets to ensure consistent contrast between structures throughout the volumes (Supplementary Figure 1b), automated segmentation of cellular structures was performed on a pixel by pixel basis (Supplementary Figure 1c) using the Ilastik software package[23] which has a user friendly interface making it easily accessible to biologists rather than requiring specific knowledge of programming languages or algorithms[24–27]. The segmentation routine was trained by manual tracing of the majority of the cellular structures of interest in the middle image of the XY, XZ, and YZ planes of each dataset and applied to the rest of the dataset. Thus, manual segmentation of <0.4% of the image volume resulted in accurate segmentation of the entire volume (Supplementary Figures 1c and 2, Supplementary Movie 1) including separation of structures within a single organelle (mitochondrial outer membrane and interior) and representing a time savings of more than 250-fold.

### Mitochondrial network scale analyses.

The cellular content, location, and, when applicable, network structure of cellular components are critical to their functional capacity. Thus, with the results of the initial segmentation, we first focused on a network scale analysis of mitochondria in the intermyofibrillar regions of glycolytic, oxidative, and cardiac muscles (Fig. 1). While there was a nearly 10-fold range in mitochondrial contents among the muscle types (Fig. 1g) as reported previously[28,29], there were also striking differences in the overall mitochondrial network arrangement (Fig. 1a–f, h). Mitochondrial networks can take three different configurations in striated muscles; glycolytic muscle mitochondrial networks are primarily oriented perpendicular to the muscle contraction axis along the I-bands (Fig. 1a, d, h), oxidative muscle mitochondria form a grid-like network comprised of nearly equivalent amounts of parallel and perpendicularly oriented mitochondria (Fig. 1b, e, h), and cardiac muscles nearly exclusively form networks parallel to the muscle contraction axis (Fig. 1c, f, h). Each of these mitochondrial networks was comprised of many adjacent mitochondria physically connected through specialized intermitochondrial junctions between the opposing outer mitochondrial membranes[19,30,31]. However, not all mitochondria within the network are physically connected to one another; instead, many subnetworks comprised of two up to hundreds of adjacent mitochondria exist. Previously, we were not able to quantify the sizes of mitochondrial subnetworks due to the time required (hundreds of hours) to manually segment these structures[20,31]. However, with the automated segmentation routine, we now show that glycolytic muscles contain small subnetworks with a volume of $2.8 \pm 0.8$ $\mu m^3$ (mean $\pm$ SE, $n = 4$), oxidative muscles contain larger subnetworks at $25.8 \pm 13.4$ $\mu m^3$ ($n = 4$), and cardiac muscles have the largest subnetworks with a volume of $244.9 \pm 116.4$ $\mu m^3$ ($n = 3$) (Fig. 1i). These mitochondrial subnetwork sizes along with the overall network orientation are consistent with functional electrical connectivity of the mitochondrial networks in striated muscle fibers. As shown in Supplementary Figure 3, oxidative muscle mitochondria appear to be similarly functionally connected along both the perpendicular and parallel axes of the muscle as evident

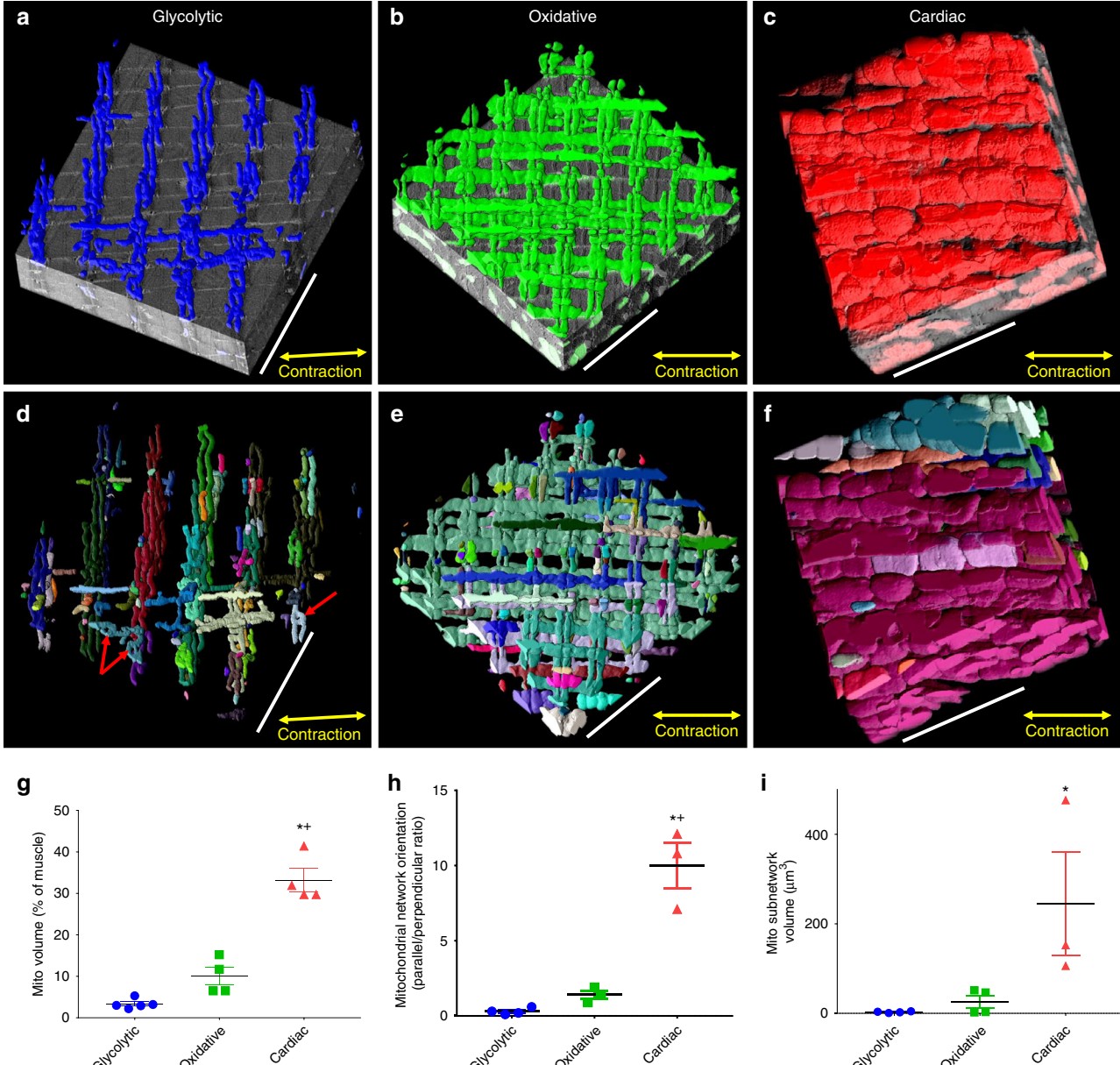

**Fig. 1** Mitochondrial network connectivity depends on muscle fiber type. **a** 3D rendering of perpendicularly oriented glycolytic muscle mitochondrial network partially overlaid on raw serial block face scanning electron microscopy (SBF-SEM) data. **b** Grid-like oxidative muscle mitochondrial network partially overlaid on raw focused ion beam scanning electron microscopy (FIB-SEM) data. **c** Parallel cardiac muscle mitochondrial network partially overlaid on raw FIB-SEM data. **d** 3D rendering of connected subnetworks within the glycolytic muscle mitochondrial network. Each color represents a connected mitochondrial subnetwork. **e** Connected subnetworks within the oxidative muscle mitochondrial network. **f** Connected subnetworks within the cardiac muscle mitochondrial network. **g** Mitochondrial volume as a percent of total muscle volume. glycolytic, $n = 5$; oxidative, $n = 4$; cardiac, $n = 4$ datasets. **h** Quantification of mitochondrial network orientation. Dotted line represents parallel equal to perpendicular. glycolytic, $n = 4$; oxidative, $n = 3$; cardiac, $n = 3$. **i** Mitochondrial subnetwork volumes. glycolytic, $n = 674$ subnetworks, 4 datasets; oxidative, $n = 1558$ subnetworks, 4 datasets; cardiac, $n = 522$ subnetworks, 3 datasets. Points are means for each dataset. Bars represent muscle type overall mean ± SE. *Significantly different from glycolytic, +significantly different from oxidative. Significance determined as $p < 0.05$ from one way ANOVA with Tukey's HSD (variances not different) or Dunn's multiple comparisons (variances different) post hoc tests. Scale bars – 5 μm

by the magnitude of the depolarization in the region adjacent to the photoinduced depolarization region. Conversely, glycolytic muscle mitochondria show greater coupling along the perpendicular axis, consistent with the physical structures, and overall lower connectivity than in the oxidative muscle which is consistent with the smaller subnetwork size in glycolytic muscle. We previously reported that cardiac muscle mitochondria have greater connectivity along the parallel axis, consistent with the orientation of the cardiac mitochondrial network[31]. Thus, the physical structures reported here appear to provide a good indicator of functional capacity.

**Mitochondrial network components analyses**. To understand the function of any network, knowledge of its core components is imperative. Thus, we next aimed to assess the structures of the individual mitochondria within each muscle mitochondrial network as a reflection of their capacity for energy distribution. However, analysis of individual mitochondria required an additional

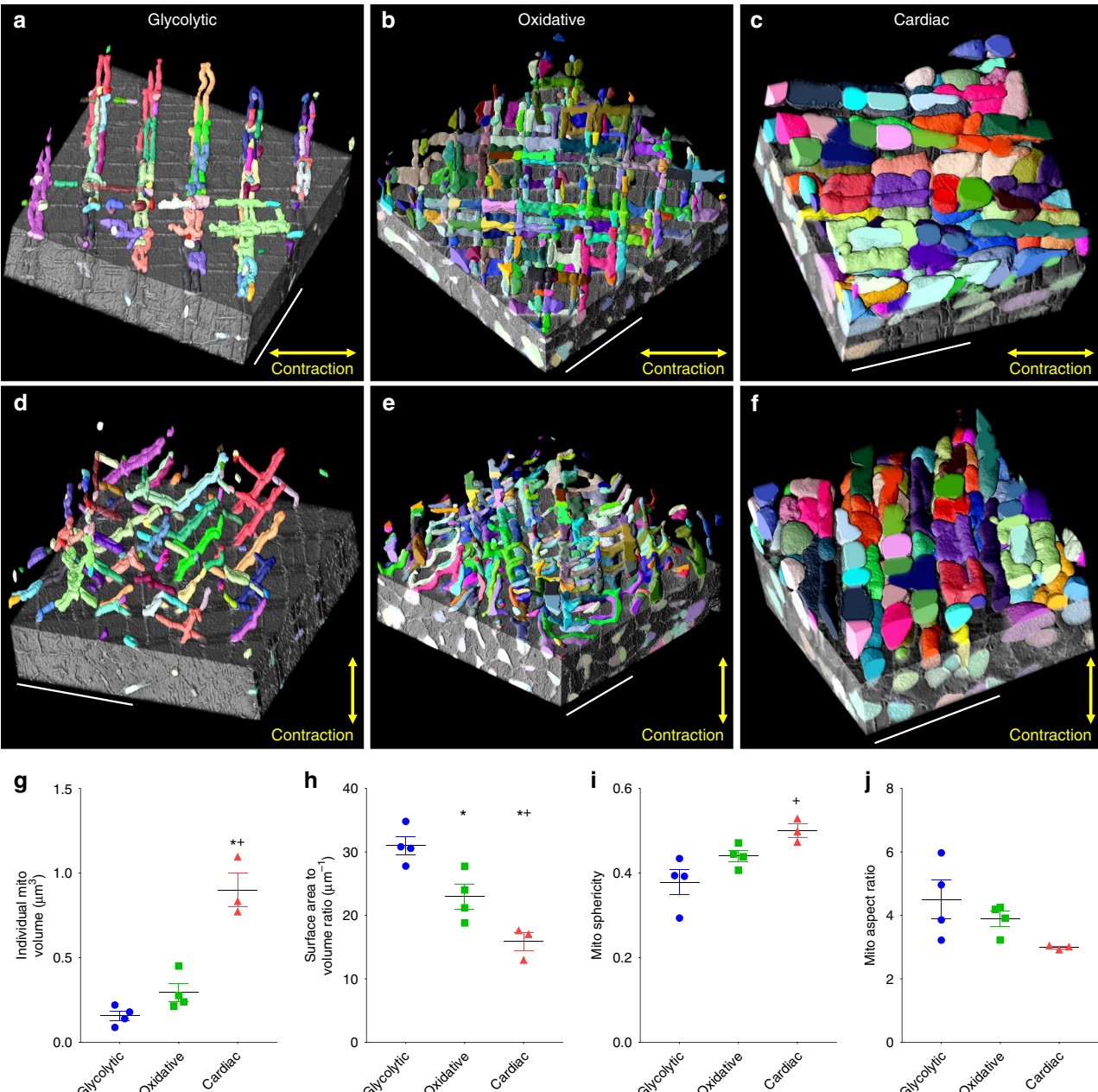

**Fig. 2** Individual network components are tuned to meet cellular goals. **a** 3D rendering of individual glycolytic muscle mitochondria partially overlaid on raw data. Each color represents an individual mitochondrion. **b** Individual oxidative muscle mitochondria. **c** Individual cardiac muscle mitochondria. **d–f** 90° rotation of **a–c**. **g** Individual mitochondrial volumes. **h** Mitochondrial surface area to volume ratios. **i** Mitochondrial sphericities. **j** Mitochondrial aspect ratios. glycolytic, $n = 1336$ mitochondria, 4 datasets; oxidative, $n = 2623$ mitochondria, 4 datasets; cardiac, $n = 1890$ mitochondria, 3 datasets. Points are means for each dataset. Bars represent muscle type overall mean ± SE. *Significantly different from glycolytic, +significantly different from oxidative. Significance determined as $p < 0.05$ from one way ANOVA with Tukey's HSD (variances not different) or Dunn's multiple comparisons (variances different) post hoc tests. Scale bars – 5 μm

segmentation step as the initial routine, similar to previous approaches[24,27], only determined where mitochondria were located within a 3D dataset but could not distinguish adjacent mitochondria from one another. Segmentation of individual mitochondria was achieved by applying the results of the initial segmentation of the outer mitochondrial membranes to the multicut[32] module in Ilastik which uses a simple user interface to determine the boundaries of cellular structures in an automated fashion (Supplementary Figure 1d). Individual mitochondrial structures were determined with ~98% accuracy (Supplementary Figure 4) and allowed for high throughput measurement and visualization of

mitochondrial size and shape (Supplementary Figure 5, Supplementary Movie 2). Mitochondrial cristae density as well as oxidative phosphorylation capacity and protein composition have been previously shown to be similar between different striated muscle types when normalized to mitochondrial volume[28,29,33–35]. Thus, the mitochondrial structures determined here likely reflect functional capacity. Cardiac mitochondria were larger than both glycolytic and oxidative muscle mitochondria (Fig. 2) reflecting a greater energy distribution capacity at the individual mitochondrial level. Conversely, glycolytic mitochondria had the greatest surface area to volume ratio suggesting a greater capacity for these mitochondria to

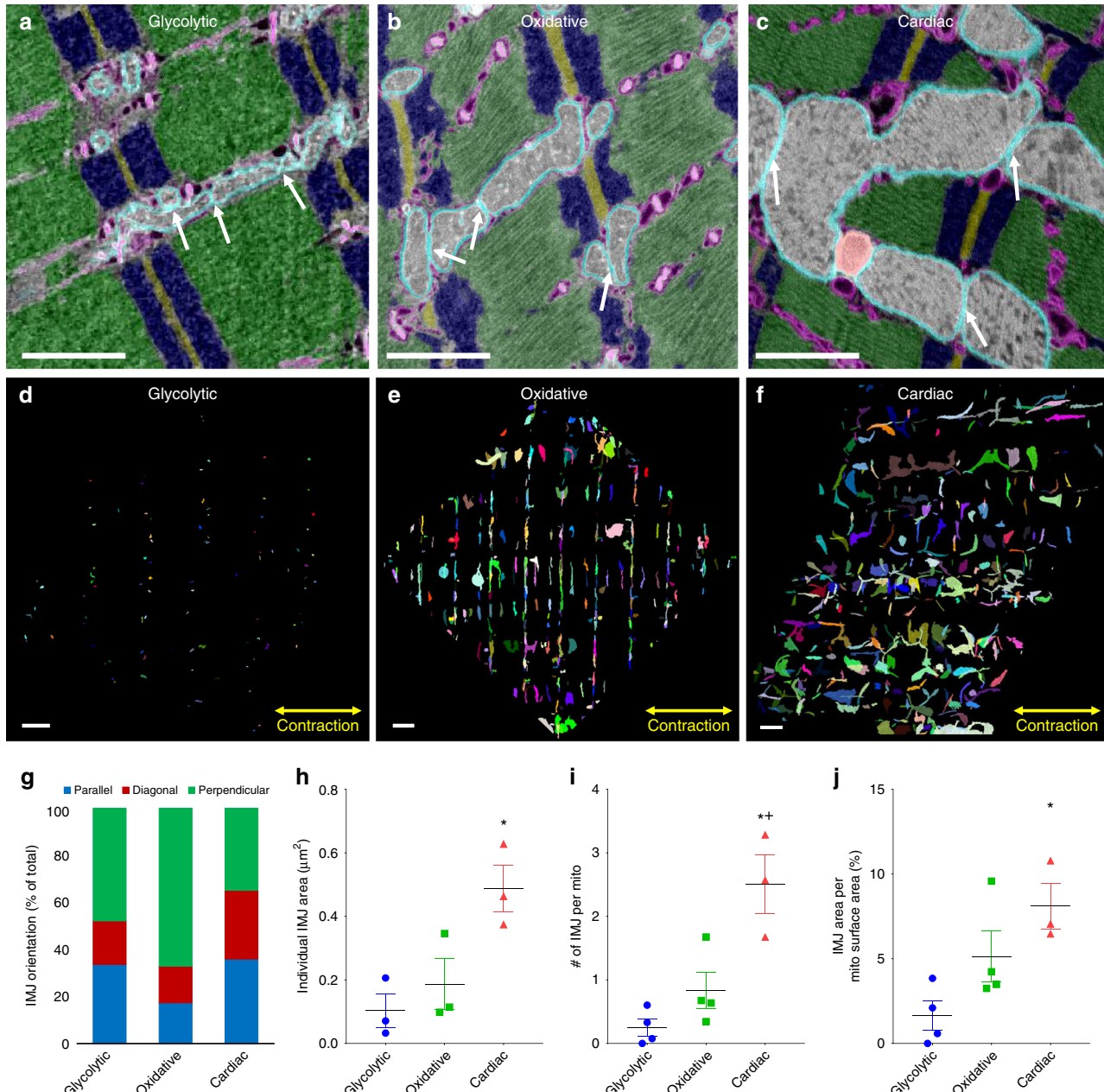

**Fig. 3** Intermitochondrial junction morphology varies by muscle type. **a** Overlay of raw and segmented single image from glycolytic muscle SBF-SEM volume highlighting intermitochondrial junctions (IMJs, white arrows). Outer mitochondrial membrane – cyan, mitochondrial interior – gray, sarcoplasmic reticulum and t-tubules – magenta, z-disks – brown, I-bands – blue, A-bands – green. **b** Segmented and raw single image from oxidative muscle FIB-SEM volume highlighting IMJs. **c** Segmented and raw single image from cardiac muscle FIB-SEM volume highlighting IMJs. **d–f** Maximum projections of all glycolytic (**d**), oxidative (**e**), and cardiac (**f**) muscle IMJs. **g** Quantification of IMJ orientation. **h** Individual IMJ sizes. glycolytic, $n = 170$ IMJs, 3 datasets; oxidative, $n = 952$ IMJs, 3 datasets; cardiac, $n = 2403$ IMJs, 3 datasets. **i** Number of IMJs per mitochondria. glycolytic, $n = 1336$ mitochondria, 4 datasets; oxidative, $n = 2623$ mitochondria, 4 datasets; cardiac, $n = 1890$ mitochondria, 3 datasets. **h** IMJ Area per mitochondrial surface area. glycolytic, $n = 1336$ mitochondria, 4 datasets; oxidative, $n = 2623$ mitochondria, 4 datasets; cardiac, $n = 1890$ mitochondria, 3 datasets. Points are means for each dataset. Bars represent muscle type overall mean ± SE. *Significantly different from glycolytic, +significantly different from oxidative. Significance determined as $p < 0.05$ from one way ANOVA with Tukey's HSD (variances not different) or Dunn's multiple comparisons (variances different) post hoc tests. Scale bars – 1 µm

interact with the surrounding environment. Indeed, glycolytic mitochondria face the highest cytosolic calcium levels during muscle contraction and have higher calcium retention capacities[36,37]. Thus, individual mitochondrial structures appear consistent with their respective functions.

**Mitochondrial network junctions analyses.** Communication and distribution across a network is often regulated by the junctions connecting the individual components. Mitochondrial networks are no different and are connected by specialized, dynamic intermitochondrial junctions (IMJs) that appear to be capable of both closing and physically separating quickly to prevent the spread of dysfunction[19,30,31] (Fig. 3a–c). To perform high throughput assessment of IMJs in the 3D muscle volumes, we segmented individual IMJs by isolating all the mitochondrial pixels that were directly adjacent (within 10 nm) to another

mitochondrion using ImageJ software (Fig. 3d–f). The size, number, and surface area in contact with IMJs per mitochondrion was highest in cardiac muscle consistent with its larger, more connected mitochondria (Fig. 3h–j). However, while the IMJs in glycolytic and cardiac muscles were oriented in all directions (Fig. 3d, f, g), the IMJs in oxidative muscle were highly ordered with $67.1 \pm 3.2\%$ ($n = 3$) oriented perpendicular to the muscle contraction axis (Fig. 3e, g). As such, though the oxidative mitochondrial network is oriented along both the parallel and perpendicular axes (Fig. 1), the connections between mitochondria within the network primarily occur between the fiber parallel segments. It was previously reported that the thinner, perpendicular I-band segments of oxidative muscle mitochondria retract into the thicker, fiber parallel segments upon damage[31]. Thus, maintaining the majority of the intermitochondrial communication sites in the portions of the network which are more likely to persist after injury may be beneficial to the muscle regeneration process. Indeed, mitochondrial signaling has been shown to be integral to the immediate repair process upon acute damage[38].

**Connectomic analyses of organelle interactions**. Closer inspection of the 3D mitochondrial structures in glycolytic muscle revealed many donut shaped regions of mitochondria with holes in the center (see red arrows in Fig. 1d). Donut mitochondria were found in each muscle type (Fig. 4a–d), however, they were much more common in glycolytic muscle (Fig. 4e). Mitochondrial donuts have previously been linked to oxidative stress[39] although the holes observed here (as small as 80 nm in diameter) are generally smaller than the resolution limits of standard light microscopy where they were observed previously. The donut holes in glycolytic muscle contained cytosol as well as sarcoplasmic reticulum running through the center (Fig. 4a, b, Supplementary Movie 3). Conversely, the few donut holes found in oxidative and cardiac mitochondria were filled only by lipid droplets (Fig. 4c, d, Supplementary Movie 4). This observation of the different interactions between mitochondria and other cellular components led us to perform a more quantitative assessment of organelle interactions within each muscle fiber type. Using the segmented cellular structures from above, we first explored mitochondrial-sarco(endo)plasmic reticulum (SR) contacts (Supplementary Figure 1e, Supplementary Movie 5) which have been implicated in calcium cycling, protein and lipid trafficking, and other critical cell processes[5,16,17]. Mitochondria-SR contact sites, defined as 30 nm or less space between the organelles, were found in 97.3, 97.5, and 97.1% of all mitochondria examined in glycolytic, oxidative, and cardiac muscle, respectively, and were highest in glycolytic muscle and lowest in cardiac muscle (Fig. 4f) consistent with the calcium cycling demands of the respective muscles[36,37]. Lipid droplets also formed intimate contacts with mitochondria in $20.4 \pm 2.4\%$ ($n = 4$) of oxidative mitochondria and $46.3 \pm 3.6\%$ ($n = 4$) of all cardiac mitochondria (Fig. 4g) providing a direct link between energy storage, conversion, and distribution sites. However, there were no lipid droplets found in glycolytic muscles consistent with the primary use of glucose in those cells. Proximity, rather than connectivity, analyses revealed that ~80% of cardiac mitochondrial volume is within 800 nm of a lipid droplet whereas that distance must double to 1600 nm to reach ~80% of oxidative mitochondria. Further, oxidative muscle mitochondria are more likely than glycolytic muscle to be in close proximity (100 nm) to myosin, the primary site of ATP utilization in contractile muscle. This greater proximity to myosin results in shorter diffusion distances for ATP and other energy metabolites between the primary sites of ATP production and utilization and is likely due to the grid-like network structure in which oxidative muscle mitochondria surround

myosin like a cage as opposed to glycolytic muscle in which the mitochondria only surround the myofibrils on a single axis.

**Organelle connectomics reveals specialized mitochondria**. We further explored the role of mitochondria-lipid droplet contacts by examining the differences between mitochondria which were and were not connected to lipids (Fig. 4j, Supplementary Movie 6). Mitochondria directly connected to lipid droplets (i.e. within 30 nm) were larger and longer than their non-connected counterparts (Fig. 4k, l) suggesting a greater capacity for ATP production as well as for energy distribution throughout the cell. However, lipid droplet-connected mitochondria had lower surface area-to-volume ratios (Fig. 4n) consistent with a reduced capacity to interact the surrounding cellular environment. Indeed, lipid droplet-connected mitochondria have fewer SR contact sites (Fig. 4o) which suggests a lower capacity for calcium cycling. These results suggest that mitochondria within muscle mitochondrial networks are specialized to perform specific cellular functions and are consistent with recent work showing structural and functional differences between cytoplasmic and lipid droplet-associated mitochondria in brown adipose tissue[40]. Mitochondria which are in direct connection with energy storage sites in muscle have larger volumes available to convert the stored fatty acids to ATP and are longer and in greater contact with adjacent mitochondria (Fig. 4m) suggesting a greater capacity to distribute energy throughout the mitochondrial network. Conversely, the lower capacity to convert and distribute energy in non-lipid droplet connected mitochondria appears to be made up for with a greater capacity for communication with both the cytosol and other organelles.

## Discussion

With the development of a connectomics framework, we were able to segment and analyze the organelle connectivity of single cells within large tissues in a high throughput manner. The power of this connectomics workflow was demonstrated through the assessment of mitochondrial structures at the network, subnetwork, individual component, and individual junction levels. Mitochondrial network configuration in striated muscles can vary from a cell with roughly one third of its volume occupied by mitochondrial aligned parallel to the contraction axis (cardiac) to a cell with 10-fold less mitochondria which are oriented along the perpendicular axis adjacent to the z-disks (glycolytic). While mitochondrial content is generally correlated with the oxidative energy demand of a muscle cell, mitochondrial network configuration may be reflective of cellular demands for contractile power. Thick mitochondria oriented parallel to the contractile axis take up space that otherwise could be used for myofibrils and lower the effective fiber cross-sectional area available for force generation. Conversely, thin mitochondrial tubules wrapping around the sarcomere near the z-disk comes at a smaller cost to myofibril cross-sectional area. Thus, there is a trade-off between maintenance of energy homeostasis and muscle power where cardiac muscle structure is balanced more for energy homeostasis than power, and, at the other end of the spectrum, glycolytic muscle structure is balanced for high power at the expense of energy homeostasis.

The overall mitochondrial network is comprised of discrete, physically separate mitochondrial subnetworks comprised of two up to hundreds of individual mitochondria. Adjacent mitochondria (i.e. within 10 nm) within a subnetwork can form specialized IMJs which have been suggested to facilitate communication and energy distribution between mitochondria[19,30,31]. Thus, it was predicted that the mitochondrial membrane potential could be propagated both along an

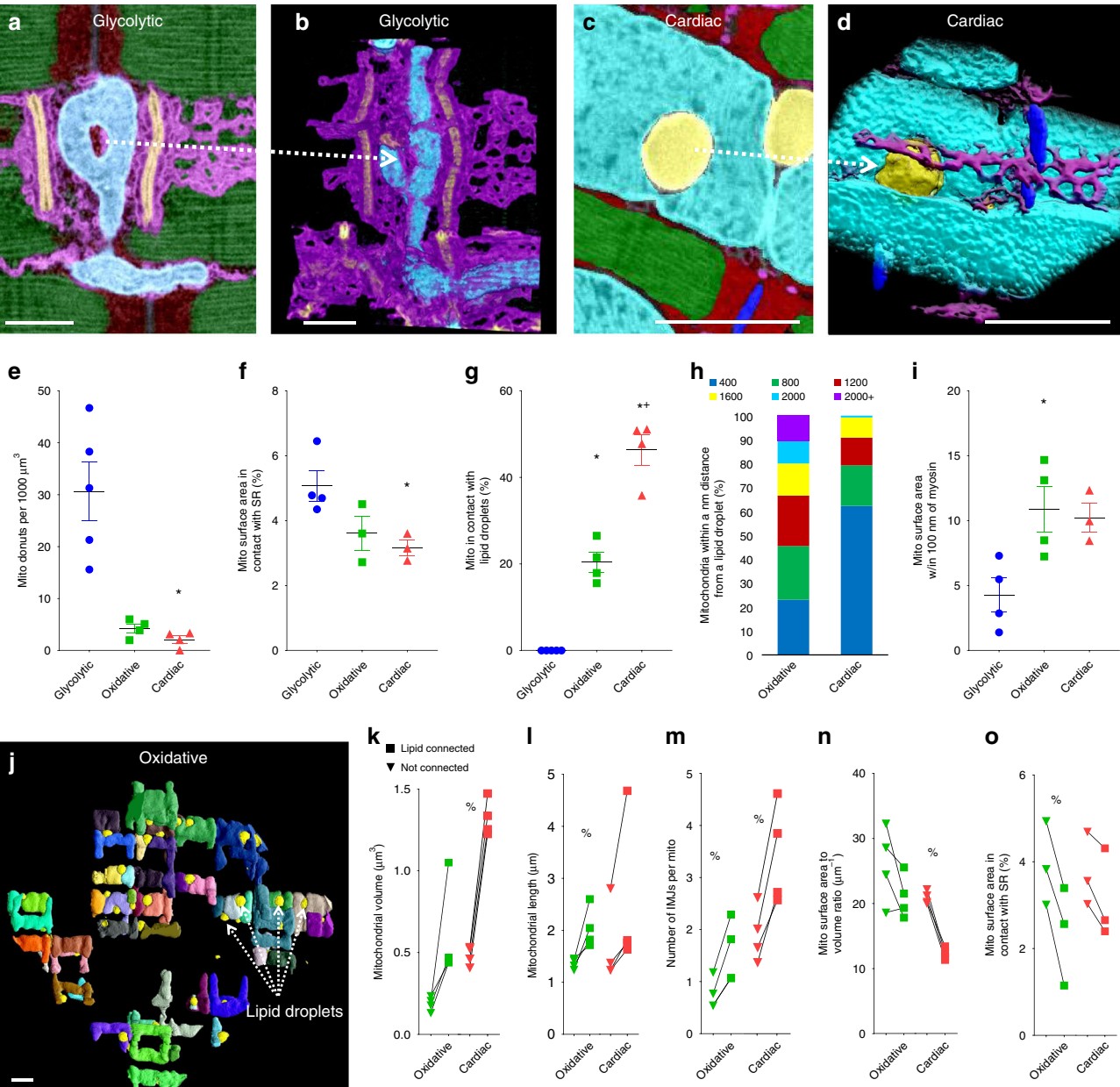

**Fig. 4** Individual mitochondria play specialized roles within the network. **a** Overlay of segmented and raw single image from glycolytic muscle FIB-SEM volume showing a mitochondrion (cyan) with a donut hole partially filled with sarcoplasmic reticulum (magenta). T-tubules – orange, z-disks – blue, I-band – red, A-band – green. **b** 3D rendering of the data in a showing only mitochondria, sarcoplasmic reticulum and t-tubules. **c** Overlay of segmented and raw single image from cardiac muscle FIB-SEM volume showing a mitochondrion with a donut hole completely filled with a lipid droplet (yellow). **d** 3D rendering of the data in c showing only mitochondria, sarcoplasmic reticulum, and lipid droplets. **e** Mitochondrial donuts per 1000 μm$^3$ muscle volume. glycolytic, $n = 144$ donuts, 5 datasets; oxidative, $n = 26$ donuts, 4 datasets; cardiac, $n = 26$ donuts, 4 datasets. **f** Mitochondrial surface area in contact with sarcoplasmic reticulum. glycolytic, $n = 637$ mitochondria, 4 datasets; oxidative, $n = 1488$ mitochondria, 3 datasets; cardiac, $n = 1019$ mitochondria, 3 datasets. **g** Percentage of mitochondria in contact with lipid droplets. glycolytic, $n = 720$ mitochondria, 5 datasets; oxidative, $n = 2410$ mitochondria, 4 datasets; cardiac, $n = 1462$ mitochondria, 4 datasets. **h** Mitochondria within given distance from a lipid droplet. **i** Percentage of mitochondrial surface area within 100 nm of myosin. $n = 637$ mitochondria, 4 datasets; oxidative, $n = 2410$ mitochondria, 4 datasets; cardiac, $n = 1019$ mitochondria, 3 datasets. **j** 3D rendering of lipid-connected mitochondria in oxidative muscle. Mitochondria – various colors, lipid droplets – yellow. **k**–**n** Mitochondrial volumes (k), surface area-to-volume ratios (l), lengths (m), and surface area in contact with sarcoplasmic reticulum (n) for lipid droplet-connected and non-connected mitochondria. oxidative, $n = 2410$ mitochondria, 4 datasets; cardiac, $n = 1462$ mitochondria, 4 datasets. Points are means for each dataset. Squares- lipid droplet connected, triangles – non-lipid droplet connected. Lines between points indicate paired values from within single dataset. Bars represent muscle type overall mean ± SE. *Significantly different from glycolytic, [+]significantly different from oxidative, [%]significantly different from lipid-connected. Significance determined as $p < 0.05$ from one way ANOVA with Tukey's HSD (variances not different) or Dunn's multiple comparisons (variances different) post hoc tests for fiber type differences, and for a paired $t$-test (normal distribution) or Wilcoxon signed rank test (not normal distribution) for lipid connected analyses. Scale bars – 1 μm

individual mitochondrion and across IMJs to adjacent mitochondria and provide a rapid mechanism for cellular energy distribution[30,41]. Indeed, we find that the muscle mitochondrial reticulum is functionally connected through the mitochondrial membrane potential[20,31] and that the magnitude and direction of functional connectivity is consistent with the size and orientation of the mitochondrial subnetworks (Supplementary Figure 3). We consistently observed functional mitochondrial connectivity at distances longer than 10 μm along both axes in oxidative muscle (Supplementary Figure 3e, f). However, of the 5849 individual mitochondria assessed in this study, only 6.1% were longer than 5 μm which suggests that both individual mitochondria as well as IMJs are acting as conductive elements within a subnetwork. If both individual mitochondria and IMJs are capable of conductivity, is there a functional difference between one big, long mitochondrion and many, small mitochondria highly connected by IMJs? One potential role of IMJs is to act as a dynamic regulator of connectivity within mitochondrial subnetworks such that dysfunctional mitochondria can be quickly isolated from the remainder of the network[31]. However, one result of smaller mitochondria with more IMJs is that a greater proportion of mitochondrial volume is consumed by the outer mitochondrial membrane[42]. While this increased outer membrane surface area to volume ratio may be advantageous for interactions with the cytosol or other organelles as discussed above, one potential disadvantage comes during removal of these mitochondria when damaged. The mitophagy process involves wrapping membranes around the surface of damaged mitochondria leading to selective degradation by lysosomes[43], and relatively more mitophagic membrane would be required to complete this process on mitochondria with greater surface area to volume ratios. Thus, though big, long mitochondria and IMJs both appear to facilitate energy distribution within muscle cells, there are likely additional factors which may dictate the balance between mitochondrial size and the number of IMJs within a given mitochondrial subnetwork.

We also extended our analysis to provide high resolution, large scale information on the organelle interactome between sites of energy storage, conversion, and utilization in addition to other cellular functions including calcium cycling. This simultaneous assessment of several organelle interactions revealed the presence of specialized mitochondria which may provide a more efficient mechanism to achieving the many functions the mitochondrial network is tasked with rather than to attempt to have all mitochondria perform all functions equally. While this single cell connectomics platform was applied to healthy mouse muscle here, it should be easily applied to pathological muscle conditions such as dystrophy, heart failure, diabetes, and aging as well as to other cell types and species provided the tissues can be prepared to achieve sufficient structural contrast. Finally, it would be interesting to apply this approach in conjunction with the improving methods to genetically encode tags for electron microscopy[44,45] in order to assess the connectivity of cellular structures which may be otherwise difficult to visualize such as microtubules or other cytoskeletal structures.

In summary, the development of a subcellular connectomics framework allowed for high throughput analyses of the intra- and inter-organelle interactions of the striated muscle mitochondrial reticulum at the cellular scale. We found that the structure and connectivity of the overall mitochondrial network as well as the architecture of individual mitochondria and intermitochondrial junctions within the network are consistent with the respective energetic demands of glycolytic, oxidative, and cardiac muscles. Assessment of the interactions between mitochondria and sites of energy storage and utilization as well as calcium cycling within the cell revealed the presence of specialized mitochondria structurally optimized for either energy distribution or calcium uptake.

As demonstrated here, subcellular connectomic analyses provide a quantitative approach to assessing the coordinated and integrative nature of cellular processes warranting the continued development of efficient analytical platforms available to biologists.

## Methods

**Mice**. All procedures were approved by the National Heart, Lung, and Blood Institute Animal Care and Use Committee and performed in accordance with the guidelines described in the Animal Care and Welfare Act (7 USC 2142 § 13). Male C57BL/6 N mice, age 2–4 months, were purchased from Taconic Farms (Germantown, NY) and fed ad libitum on a 12-h light, 12-h dark cycle at 20–26 °C.

**Sample preparation**. Mice were anesthetized with 2% isoflurane via nose cone while lying on a heated bed. For skeletal muscle fixation, hindlimb skin was peeled back and hindlimbs were immersed in fixative solution (2% glutaraldehyde in 0.1 M phosphate buffer, pH 7.2) for 30 minutes in vivo. For cardiac fixation, the chest cavity was opened, and the heart was perfusion fixed with a syringe attached to a 30 G needle through the apex of the left ventricle slowly pushing 2 ml of relaxing buffer (80 mM potassium acetate, 10 mM potassium phosphate, 5 mM EGTA, pH 7.2) followed by 2 ml of fixative solution. After initial fixation, the tissues were excised, cut into 1 mm³ cubes, and placed into standard fixative solution (2.5% Glutaraldehyde, 1% Paraformaldehyde, 0.12 M sodium cacodylate, pH 7.2–7.4) for 1 h.

Samples were post-fixed and stained en bloc using an established protocol with minor modifications[46,47]. After five washes (always 3 min each) with 0.1 M cacodylate buffer at room temperature, samples were post-fixed in reduced 4% osmium solution (3% potassium ferrocyanide, 0.2 M cacodylate, 4% aqueous osmium) for 1 h on ice, washed five times in bi-distilled $H_2O$, and incubated in fresh thiocarbohydrazide solution for 20 min at room temperature. In a second post-fixation step, samples were incubated in 2% osmium solution for 30 min on ice and washed five times in bi-distilled $H_2O$. The sample was then incubated in 1% uranyl acetate solution and left in a refrigerator (4 °C) overnight, washed five times in bi-distilled $H_2O$, incubated at 60 °C for 20 min with Walton's lead aspartate (0.02 M lead nitrate, 0.03 M aspartic acid, pH 5.5), and washed five times in bi-distilled $H_2O$ at room temperature. The sample was next dehydrated in a graded ethanol series (20%, 50%, 70%, 90%, 95%, 100%, and 100%; 5 min each), and they were incubated in 50% Epon (50% ethanol) for 4 h and incubated in 75% Epon resin (25% ethanol) at room temperature overnight. Next day samples were incubated in fresh 100% Epon resin in one, one, and 4 h in order. After removing excess resin using filter paper, the Samples were placed on aluminum Zeiss SEM Mounts (Electron Microscopy Sciences, #75510) and polymerized in a 60 °C oven for 2 days. After polymerization, stubs were mounted in a Leica UCT Ultramicrotome (Leica Microsystems Inc., USA) and faced with a Trimtool 45 diamond knife (DiATOME, Switzerland) with a feed of 100 nm at a rate of 80 mm/s.

**FIB-SEM imaging**. The images were acquired with a ZEISS Crossbeam 540 using the ZEISS Atlas 5 software (Carl Zeiss Microscopy GmbH, Jena, Germany) and were collected using an In-Column Energy Selective Backscatter (ESB) with filtering grid to reject unwanted secondary electrons as well as backscatter electrons up to a voltage of 1.5 kV at the working distance of 5.01 mm. The milling was performed with a FIB operating at 30 kV and 2–2.5 nA beam current. The thickness of the FIB slices was 5–10 nm. Image stacks were aligned with a proprietary algorithm by using the Atlas 5 software (Fibics) and exported as TIFF format for further analysis.

**SBF-SEM imaging**. The resin-embedded block of muscle was gold sputter-coated (40 nm) and imaged in high vacuum with a Zeiss Sigma VP SEM (Zeiss Microscopy, USA) equipped with a Gatan 3-View SBF system (Gatan Inc., USA). All image stacks were acquired at 1.4 kV. Each image in the stack was 2000 × 2000 pixels acquired with 1 μs/pixel dwell time at an electron dose of ~25 e/nm². 300 slices were acquired with a pixel size of 6 nm in x, y and 35 nm in z. All image stacks were then aligned to correct for x-y drift, using Digital Micrograph (Gatan Inc., USA).

**Data preparation for image segmentation**. All image processing was performed on a desktop PC (Thinkmate, Waltham, MA) running Windows 7 with an Intel Xeon E5-2670 2.6 GHz processor and 64 GB of RAM. Prior to segmentation, aligned FIB-SEM volumes were binned to 10–20 nm isotropic voxels while the aligned SBF-SEM volume was binned to 12 × 12 × 35 nm anisotropic voxels to reduce file size and therefore computational time while still maintaining sufficient resolution to discriminate cellular structures. Variable contrast throughout the datasets was then normalized using the Enhance Local Contrast (CLAHE3Dstack) tool in ImageJ (National Institutes of Health, Bethesda, MD, ImageJ.net) when necessary.

**Segmentation of cellular structures**. Initial segmentation of cellular structures was performed using the Pixel Classification module in the Ilastik software package[23] (Ilastik.org). Raw data was loaded as up to 500 sequential $1000 \times 1000$ pixel, 8-bit TIF images. Features used to train the pixel classifier were: Color/Intensity 0.3–10 pixels, Edge 0.7–1.6 pixels, and Texture 3.5–10 pixels. The pixel classifier was trained by tracing cellular structures in the middle XY, XZ, and YZ planes of the volume. For mitochondria-specific analyses, training labels were created for the mitochondrial outer membrane, mitochondrial interior, and extramitochondrial pixels. For analyses involving interactions between different cellular structures, training labels were created for the mitochondrial outer membrane, mitochondrial interior, lipid droplets, sarcoplasmic reticulum, t-tubules, and the sarcomeric I-bands, A-bands, and Z-disks when visible. After initial training, the Live Update tool was used to display the results of the initial segmentation and check for mislabeling errors. When necessary, additional training iterations were performed to minimize errors. Upon satisfactory segmentation results, the pixel probability maps were exported as 32-bit HDF5 files.

To segment individual mitochondria, additional training to detect the boundaries between adjacent mitochondria was necessary. This secondary step was performed using the MultiCut[32] module in Ilastik. Raw data was loaded just as for pixel classification and then the HDF5 file containing the outer mitochondrial membrane probabilities was loaded. Superpixels were then created using the outer membrane probabilities as the input channel and default other settings. Superpixels were examined to ensure they followed the boundaries of each mitochondrion, and if not, the threshold was adjusted and the superpixel watershed updated. Generally, proper training during the pixel classification step ensures the successful creation of appropriate superpixels. Training on the superpixel boundaries was done by marking mitochondrial boundaries red and non-mitochondrial boundaries green and selecting the means of the raw data and outer membrane probabilities standard (edge) and standard (sp) as features. The Live Predict tool was used to determine how much training was necessary (usually ~15 minutes) to get accurate predictions before using the Live Multicut tool with the Nifty_FmGreedy solver and 0.3 beta to generate the initial individual mitochondrial segmentations. Increased accuracy was obtained by additional boundary training or adjusting the beta value (lower for more merging of objects, higher for more splitting) and updating the multicut when necessary. Upon satisfactory results, the multicut segmentation was exported as a 32-bit HDF5 file for analysis. In general, it took 3.5–4 h to complete the data preparation, initial pixel classification, and individual mitochondrial segmentation routines including all data loading, training, computation, and export.

**Segmentation accuracy**. All image analysis was performed in ImageJ. To determine segmentation accuracy, an overlay of the individual mitochondrial multicut segmentation and the raw data (50 sequential $800 \times 400$ pixel images or 16 μm[3] from the middle of a dataset) was loaded into the TrakEM2 plugin[48]. Accuracy was assessed in each image on a pixel by pixel basis by marking all pixels that were assigned to a mitochondrion when they should not have been and all pixels that were not assigned to a mitochondrion when they should have been. Accuracy was also assessed at the mitochondrial level by counting all mitochondria in each raw image as well as each mitochondrion that was improperly merged with another, improperly split, or missed entirely.

**Mitochondrial network analysis**. All HDF5 files created in Ilastik were loaded into ImageJ using the Ilastik Import from HDF5 plugin. To examine overall mitochondrial network parameters, an individual mitochondrial segmentation (multicut) dataset was binarized to select all mitochondria. Total mitochondrial volume was determined as the percentage of binarized mitochondrial pixels per total muscle fiber pixels. Overall mitochondrial network orientation was determined by first rotating the binarized mitochondrial dataset in 3D until the muscle contraction axis was horizontally aligned. The proper rotation was first determined using the raw data where the regular sarcomeric structure could be visualized and aligned such that the z-disks were vertically aligned in both the XY and XZ planes using the Rotate and Reslice tools, and then the same rotations were applied to the mitochondrial dataset. The average projection image of both the XY and XZ planes of the mitochondrial image were taken using the Z Project tool, binarized using the Threshold tool, and the OrientationJ Distributions[49] plugin was used to determine the angles of the mitochondrial network. Parallel mitochondria were determined as those with a $\pm 0$–$10°$ angle and perpendicular mitochondria were determined as those with a $\pm 80$–$90°$ angle. Mitochondrial subnetworks were determined by using the Find Connected Regions plugin on the binarized mitochondrial datasets and volumes were determined as the number of pixels using the known pixel volumes. The number of mitochondria per subnetwork was determined by first dividing the 32-bit multicut mitochondrial segmentation dataset by the number of mitochondria within the dataset using the Divide tool and adding the resultant image to the connected regions image created above using the Image Calculator tool while choosing 32-bit (float) result. Then a histogram was created with the number of bins equal to the number of mitochondria multiplied by the number of created regions. The number of different values found within each connected region value was then counted as the number of mitochondria within each subnetwork. To better represent the average mitochondrion within a muscle volume and reduce the influence of several small subnetworks in the presence of one very large

subnetwork, the subnetwork volumes and number of mitochondria per subnetwork are reported as the volume weighted averages.

**Individual mitochondrial analysis**. The 32-bit multicut mitochondrial segmentation datasets were converted to 16-bit images with the background given a 0 value, and individual mitochondrial volumes and surface areas were measured using the 3D Geometrical Measure tool in the ROIManager3D plugin[50]. Individual mitochondrial sphericity, or 3D form factor, elongation (largest radius/middle radius), and flatness (middle radius/smallest radius) were measured using the 3D Shape Measure tool in the ROIManager3D plugin. Mitochondrial aspect ratio (largest radius/smallest radius) was calculated by multiplying the elongation and flatness values together. Mitochondrial length was measured using the Geodesic Diameter 3D tool in the MorphoLibJ plugin[51]. Visual inspection of measured mitochondrial morphological values was performed using the Assign Measure to Label tool in the MorphoLibJ plugin after importing a Results table containing the mitochondrial label numbers and matching morphological values. All 3D visualizations were done using the 3D Viewer or Volume Viewer plugins. Mitochondrial donuts were defined as any portion of a mitochondrion with a hole completely through the middle and were counted by visual inspection of all the mitochondria within a dataset using the 3D Viewer plugin.

**Intermitochondrial junction segmentation**. Intermitochondrial junctions (IMJs) were defined as the regions where pixels from two different mitochondria were directly adjacent to one another. First, a mitochondrial mask with mitochondrial pixels assigned a value of 1.0 and background pixels assigned as null (NaN or ignored in all mathematical analyses) values was created by dividing the binarized all mitochondria image described above by itself using the Image Calculator tool while choosing 32-bit (float) result. The mitochondrial mask image was multiplied by the 16-bit individual mitochondrial segmentation image using the Image Calculator tool while choosing 32-bit (float) result. A $1 \times 1 \times 1$ pixel Mean 3D filter was then performed resulting in an image where only pixels directly adjacent to another mitochondrion would change value. A difference image between the 3D mean filtered image and the 16-bit individual mitochondrial segmentation image was then calculated using the Image Calculator tool while choosing 32-bit (float) result. The difference image was then divided by itself using the Image Calculator tool and the multiplied by 65535 using the Multiply tool yielding a final binarized IMJ image where IMJs have a value of 65535 and all other pixels are null.

**Intermitochondrial junctions analysis**. IMJ orientation was determined similarly to mitochondrial network orientation described above. The binarized IMJ image was converted to 8-bits and rotated to align the muscle contraction axis horizontally. Then a maximum projection image was created using the Z Project tool and the OrientationJ Distributions plugin was used to determine IMJ angles. IMJs oriented $\pm 0$–$29°$ were considered parallel, $\pm 30$–$59°$ diagonal, and $\pm 60$–$90°$ perpendicular. Individual IMJ area was calculated by using the Find Connected Regions plugin to determine the number of pixels in each IMJ. The number of IMJs per mitochondrion was calculated using the Region Adjacency Graph tool in the MorphoLibJ plugin to determine how many contacts each mitochondrion made with another. IMJ area per mitochondrial surface area was determined by first subtracting the 16-bit binarized IMJ image described above from the 16-bit individual mitochondrial segmentation image using the Image Calculator tool and then running the 3D Geometrical Measure tool in the ROIManager3D plugin to determine the resultant individual mitochondrial pixel volumes. IMJ area was then calculated from the difference between the IMJ subtracted mitochondrial pixel volume and the original mitochondrial pixel volumes measured above.

**Mitochondrial network interactions**. To determine the distance between mitochondria and other cellular structures, first a mask of the outer mitochondrial membranes which was created by using the Label Boundaries tool in the MorphoLibJ plugin on the 16-bit individual mitochondrial segmentation image created above. The resultant image was divided by itself using the Image Calculator tool and selecting 32-bit (float) result yielding a mask image with a value of 1.0 for all mitochondrial outer membrane pixels and null for all others. The segmented probability maps for each cellular structure were then overlaid on the raw data to determine the optimal threshold (usually ~0.5) to create binary images for each cellular structure. For each binarized cellular structure image, the Distance Transform 3D plugin was used to create a map of the distances from each cellular structure. The distance map was then multiplied by the mitochondrial outer membrane mask using the Image Calculator tool to give the mitochondrial distances from a given cellular structure. This image was then thresholded to select all values of 30 nm or less to determine contact sites with lipid droplets or sarcoplasmic reticulum and thresholded to select values of 100 nm or less to determine mitochondria within 100 nm of myosin. The thresholded images were multiplied by 65535 and then subtracted from the 16-bit individual mitochondrial segmentation image using the Image Calculator tool. The 3D Geometrical Measure tool in the ROIManager3D plugin was then used to determine the resultant individual mitochondrial pixel volumes. The percentage of the mitochondrial surface area interacting with a given cellular structure was then calculated from the difference

between the interaction subtracted mitochondrial pixel volume and the original mitochondrial pixel volumes measured above.

**Functional connectivity testing**. Mouse soleus (oxidative) and flexor digitorum brevis (glycolytic) muscle fibers were dissected and placed in an incubation medium (IM) composed of: (mM): NaHEPES (10), NaCl (137), KCl (5.4), CaCl2 (1.8), MgCl2 (0.5), NaH2PO4 (0.5), Glucose (10), NaPyruvate (1), and Butanedione monoxomine (BDM, 20), pH 7.4 plus 3 mg ml$^{-1}$ collagenase D (Roche) for 30–45 min in a shaking water bath at 37 °C. The tissue was removed from the collagenase solution and placed in the IM solution alone. The fibers were gently teased from the muscle bundle and long, non-contracted fibers were placed in a glass bottom dish with IM plus 5 nM tetramethylrhodamine methyl ester (TMRM) and 20 μM MitoPhotoDNP[52] for at least 20 min prior to imaging. Confocal microscopy and MitoPhotoDNP photoactivation of isolated fibers was performed on an inverted Zeiss 780 microscope equipped with a 355 nm laser. Mitochondrial functional connectivity was analyzed from averages of five images (~1–2 s) taken before (Pre) and immediately after UV exposure (Post). Mitochondrial and cytosolic/nuclei TMRM signals were separated for analysis by intensity thresholding each image. Ratiometric images were created by dividing the Post images by the Pre image. Mitochondrial and cytosolic/nuclei Post/Pre TMRM signals were measured as a function of distance outside the irradiated region using Plot Profile tool in ImageJ.

**Statistics**. Morphological differences between the three muscle fiber types were assessed on the mean values for each dataset by ANOVA. Brown-Forsythe tests were performed to determine the equality of group variances. When the variances were determined to be different, a Dunn's multiple comparisons test was performed. When variances were determined to not be different, a Tukey's HSD post hoc test was performed. Differences between lipid droplet-connected and not connected mitochondria were evaluated on the means from each dataset using a two-sided paired $t$-test for all data in which passed a Shapiro-Wilks normality test. For lipid connected and not connected datasets which did not pass the Shapiro-Wilks test, a Wilcoxon signed rank test was performed to compare means. Differences between glycolytic perpendicular and parallel coupling TMRM ratios were evaluated using a two-sided, equal variance $t$-test at 0, 1, 3, 5, 7, and 9 μm from UV irradiation. All tests used a $p$-value of 0.05.

## Data availability

All raw data used in this work are available upon reasonable request. Further information on the experimental design is in the linked Reporting Summary available as a Supplementary Information file.

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

## Acknowledgements

We would like to thank Erin Stempinski of the NHLBI Electron Microscopy Core for assistance with the FIB-SEM data collection and the Electron Microscopy Unit in the Trans-NIH Shared Resource on Biomedical Engineering and Physical Science as well as the Cellular and Supramolecular Structure and Function Section of the Laboratory of Cellular Imaging and Macromolecular Biophysics within NIBIB for the collection of the SBF-SEM images. This work was supported by the Division of Intramural Research of the National Heart Lung and Blood Institute and the Intramural Research Program of the National Institute of Arthritis and Musculoskeletal and Skin Diseases.

## Author contributions

C.K.E.B. and B.G. designed the imaging experiments. B.G. and Y.K. performed the in vivo tissue fixation. C.K.E.B. and Y.K. collected FIB-SEM datasets. B.G. designed and performed the image segmentation and analysis. B.G. and T.B.W. performed and analyzed the isolated muscle fiber experiments. B.G. wrote the manuscript. B.G., C.K.E.B., T.B.W., and Y.K. edited the manuscript.

## Additional information

**Competing interests:** The authors declare no competing interests.

