## [Peer Review File · Nature Communications]

Reviewers' comments:

Reviewer #1 (Remarks to the Author):

The manuscript lays down the technical foundation for organelle connectomics by devising fully automated analysis of 3D electron microscopic image stacks. The corresponding author has previously revealed muscle mitochondrial network using laborious, manual image processing. In this work, the automated pipeline allowed large-scale, quantitative characterization of the mitochondrial network down to individual mitochondria level as well as inter-mitochondrial junctions and inter-organelle contacts for three functionally different tissues of cardiac, oxidative and glycolytic muscles. The study beautifully demonstrates how differently cells shape, distribute and connect multifunctional organelles in response to different demands of the tissue.

The fully automated segmentation and quantification process using open-source softwares is applicable to other organelle connectomic studies, therefore of high interest to cell biologists. The analysis pipeline is carefully designed and described in detail. The results are beautifully visualized and statistically sound. Despite the technical soundness, I am worried about weak links in interpreting the structures in functional context, which I explained in detail in the "Major Points". I recommend the publication of the manuscript after the authors strengthens the structure-function relationships.

Major Points:

1. It is unclear which of many structural features obtained in the study is critical to the energy distribution. The candidate features include the arrangement of the mitochondrial network or subnetwork and/or intermitochondrial junction (IMJ). Especially, some of these features indicate different electrical connection. For instance, the mitochondrial network in oxidative muscle lies in two directions while the IMJs were highly ordered perpendicular to the contraction axis. Therefore, it is unclear whether the electric connectivity in oxidative muscle is in two directions or focused in perpendicular direction.

To clarify the issue, I suggest the authors to perform fluorescence imaging with membrane potential sensor such as TMRM as in Ref. 20 and Ref. 30. Especially, longitudinal vs. vertical coupling as in Fig. 3 in Ref. 30 will help answering the directionality of electric connection in oxidative muscle.

2. It is also unclear which of structural features is linked to spreading of dysfunction. The overall network, subnetwork, IMJ and individual mitochondria can contribute in different ways. I suggest the authors to perform the experiments in Fig.5 in Ref. 30.

3. In page 6, 8 and 9, the author stated that the "larger and longer" mitochondria reflect a greater ATP production and energy distribution capacity at the individual mitochondrial level. Is there any supporting data or previous works? In my opinion, these features are unlikely to correlate directly with the functions. It is because the size and length of mitochondrial outer membrane is not linearly correlated to the area of inner membrane. Since the inner membrane is where oxidative phosphorylation occurs and the membrane potential is applied, the total area of inner membrane may be better correlated to the energy production and distribution capacity than the overall size of the organelle outline. For instance, apoptotic mitochondria are large and dilated while the inner membrane is unfolded and the membrane potential is lowered. Thus, the length may be indicative of high membrane potential, but the volume may be oppositely correlated to the electrical state.

I suggest to quantify the total area of inner membrane per mitochondria using their segmented mitochondrial interior in Supp. Fig. 2c. Alternatively, if the authors show that the density of inner membrane is uniform across the samples, it will become reasonable that the interior area is correlated to the size and length of the organelle.

4. "Mitochondria-SR contact site" is of high interest in cell biology due to its multiple important actions of calcium cycling, lipid trafficking and mitochondrial fission. Because calcium ion diffusion is so fast, the contact site is usually defined as <30 nm gap between the two organelles. However, I can't find the definition of the contact sites in the text and method. Please specify the definition. Also, using the agreeable definition, segment the contact sites as in IMJ and report the patterns and dimensions.

In the manuscript, "proximity" is used to compare the contact sites in the three muscles. However, the average proximities of hundreds of nanometers in page 8 are unlikely correlated to the functional contacts because only <30 nm proximity is functional.

5. The manuscript ends with the suggestion that specialized mitochondria (i.e. longer, larger mitochondria contacting with lipid droplet have greater capacity for energy conversion and distribution while non-lipid droplet connected mitochondria has greater capacity for calcium cycling) as mechanism for achieving tailored functions of mitochondrial network. To support the statement, there should be supports from previous publications, or functional measurements of membrane potential and dysfunction as in my points #1 and 2, or more relevant analysis of existing data set as in my points #3 and 4.

Minor Comments:

1. Ref. 20 also claims "automated segmentation". Compare the current method to the previous ones explicitly, rather than simply referring to the old one as "manual segmentation".

2. Page 4. 1st paragraph, last sentence. "... representing a time savings of more than 250-fold". 250-fold comparing to what?

3. Page 5. Add definition of "subnetwork".

4. Page 5. For the quantified number of mitochondria per subnetwork, specify the number of subnetwork or total mitochondria to show the sample size.

5. Page 7. Last sentence. "the few donut holes found in oxidative and cardiac mitochondria were filled completely by lipid droplets". What is the total number of donut holes counted? "Completely" is not statistically meaningful without the same size.

6. Page 9. "... consistent with the calcium cycling demands of the respective muscles". Give references.

7. Page 9. "mitochondria-lipid droplet contacts". What is the definition used in this work?

8. Page 10. "... genetically encode tags for electron microscopy in order to assess individual protein interactions with cellular structures at this scale."

The EM tags involve with radicals of small molecules for increasing local electron density. Since the diffusion speed of small molecule is quite high, the radical spreads out tens of nanometers, which is far larger than molecular scale. Also, each tag produces multiple radicals, so distinguishing individual protein is near impossible. Immunogold EM or super-resolution fluorescence microscopy may be better suited for addressing individual protein interactions.

Reviewer #2 (Remarks to the Author):

In this manuscript, authors present an automated segmentation approach to separate all visually discernable components within the cell. They validate their approach in three muscle cell types and show the difference in mitochondrial network and connectivity in these different cell types.

The manuscript presents significant methodological advance that will be clearly valuable for 3D EM. The limitation of the manuscript is that it is limited to development of new methodological approach and do not present scientific advance beyond that.

Comments:

Tukey's post hoc test and t-test assume a Gaussian distribution. Authors do not mention whether their data was tested for normal distribution.

N is very low for cardiac samples (n=2 as mentioned in Figure 1 and 4) and it seems that also that all other measurements are made only from 2 cardiac samples. Low sample number does not allow to take into account the sample to sample variability.

Numerical data is presented only in bar charts. Authors should show the data points using either scatter charts or supplemental excel files.

Reviewer #3 (Remarks to the Author):

Bleck and coauthors in "Connectomics within a Single Cell: Energy Networks in Striated Muscle" present a morphological description and comparison of mitochondrial "networks" in three different striated muscle types and an improved version of technology they previously used to study mitochondrial structure with electron microscopy. Altogether, I do not recommend this paper to be published in Nature Communications because of methodical shortcomings, data overinterpretation and marginal novelty.

#1

A critical problem with the presentation and interpretation of data is that the authors interpret any mitochondrial outer membrane apposition in between adjacent mitochondria as electrically conductive, syncytium forming connection. There is no published or currently presented data supporting this. The existence of intermitochondrial junctions (IMJs) is known for about 4 decades, but their functional role, and whether or not they are able to mediate, or are involved in forming electric networks of mitochondria remains unknown. This includes previous work of the authors (Glancy et al., 2015, 2017) that shows a limited spread of photo-induced depolarization of mitochondria indicating a level of limited mitochondrion to mitochondrion communication. Importantly, this cannot prove the involvement of IMJs, because apposed and lumenally continuous mitochondria are not discernible in optical microscopy, plus the authors did not control for uncaged DNP diffusion or localized damage by the uncaging chemistry in these earlier works that may provide an alternative explanation for the observations. Thus electric conduction through IMJs remains a speculation, as the authors noted in their earlier papers: "However, the precise physical nature as well as the frequency and conductivity of these specialized IMJs remain unclear." In Glancy et al 2017 and "We speculate that the EDCS are also conductive elements specifically within the PVM and FPM pools." In Glancy et al 2015. Speaking of the currently reviewed paper, therefore the functional interpretation of structural findings is merely speculative.

#2

The novelty of the manuscript is marginal. The methodology is not novel, FIB/SEM and SBF/SEM has been used before to visualize mitochondria (including the authors), and also automated segmentation of mitochondria (Márquez Neila et al., 2016). The authors did not develop but applied and optimized a publicly available technology. The comparison of three different muscle tissue goes beyond previous published data from the authors in terms of detail level of description of mitochondrial network and morphology, but in general not a top tier novelty.

#3

Statistics is unclear and possibly wrong. For what are the n-values? Some n-values are very high (100-1000) suggesting that mean \pm SE values were calculated per mitochondrion basis. It is clearly incorrect to use populations of individual mitochondria to draw conclusions on comparing different preparations. Systematic errors, e.g. fixation, sample handling and imaging artifacts will

affect the entire population. N-values and respective standard errors must reflect independent experiments. Putting n=100 mitochondria in a statistical comparison will result in a false confidence corresponding to 100 experimental replicates (e.g. using 100 mice)!

#4

Some comparisons were made using different imaging techniques. E.g. in Fig 1 glycolytic muscle are from SBF-SEM and other from FIB-SEM. Can this bias the comparison of differing tissues?

#5

The conditions of fixation were very different between the heart and the skeletal muscle. This potentially confounds any conclusion on comparison of these tissues. E.g. Ca²⁺ was chelated with EGTA in the heart before perfusion fixation, while skeletal muscle was fixed in whole. Could observed donut shaped mitochondria (w/o lipid droplet) be due to fixation artifacts?

#6

Imprecise or lacking definitions

- Pg 3, line 8 from bottom: "connectomics approach" (and "connectomics framework" on pg 9 middle): please explain what it means. SFig 1 describes mostly image analysis, and "Network Connectivity and Components Analysis" remains unexplained.
- Pg 5, Ln 10-14: mitochondrial networks, physical connection and intermitochondrial junctions need to be more precisely defined. These definitions are murky and confusingly used through the manuscript. It should be clear where do authors indicate luminal continuity vs. apposition of mitochondrial outer membranes without luminal continuity. The current study is purely structural, so no inferences can be made to electrical connectivity. Based on previous works from the authors and others, apposition of mito membranes is not equal to IMJ or electrical connectivity. I suggest to create a distinct term for the observed "mitochondrial networks" that comprise of packed, but not lumenally continuous mitochondria, that reflects the uncertainty of electrical connectivity.

#7

Speculative overinterpretation of the data is frequent through the paper. Observed morphological differences of mitochondria (defined by the outer membrane), without the knowledge of inner membrane structure or protein expression are associated with a wide range of function:

- Abstract, "Recently, electrical connectivity within muscle mitochondrial networks was demonstrated to provide a rapid mechanism for cellular energy distribution." – this is an overstatement of the previous results of the authors. Based on those data, some level of electrical connectivity can be concluded, but cannot be stated in a generalized manner.
- Pg. 6. Ln 7 from bottom (and pg 9 first paragraph): "suggesting a greater capacity" transporters specific to metabolites and ions are in the mitochondrial inner membrane. To suggest a greater capacity for interaction, the knowledge of surface of the inner membrane, density of cristae junctions and the surface density of relevant carrier proteins would be required. The authors overinterpret the data.
- Pg 7 Ln 10-11 from below: "priority", "higher dynamic capacity" what do these mean, and how are they concluded from data? This statement sounds like an overinterpretation of the author's previous data.

Minor points:

The "null" floating point pixel value may require some explanation for lay reader (to not to confuse with zero).

Pg 17 Ln 13 Please clarify the filter used. Likely you used some kind of maximum or morphological dilation filter, but not an averaging filter. I don't expect an averaging filter resulting in the described changes in pixel values.

Pg 18 top: IMJs are surfaces. How were orientation angles defined? In a projection?

References

Glancy, B. et al. (2015) 'Mitochondrial reticulum for cellular energy distribution in muscle.', *Nature*, 523(7562), pp. 617–20. doi: 10.1038/nature14614.

Glancy, B. et al. (2017) 'Power Grid Protection of the Muscle Mitochondrial Reticulum.', *Cell reports*. ElsevierCompany., 19(3), pp. 487–496. doi: 10.1016/j.celrep.2017.03.063.

Márquez Neila, P. et al. (2016) 'A Fast Method for the Segmentation of Synaptic Junctions and Mitochondria in Serial Electron Microscopic Images of the Brain', *Neuroinformatics*, 14(2), pp. 235–250. doi: 10.1007/s12021-015-9288-z.

Response to Reviewers

Reviewers' comments:

Reviewer #1 (Remarks to the Author):

The manuscript lays down the technical foundation for organelle connectomics by devising fully automated analysis of 3D electron microscopic image stacks. The corresponding author has previously revealed muscle mitochondrial network using laborious, manual image processing. In this work, the automated pipeline allowed large-scale, quantitative characterization of the mitochondrial network down to individual mitochondria level as well as inter-mitochondrial junctions and inter-organellar contacts for three functionally different tissues of cardiac, oxidative and glycolytic muscles. The study beautifully demonstrates how differently cells shape, distribute and connect multifunctional organelles in response to different demands of the tissue.

The fully automated segmentation and quantification process using open-source softwares is applicable to other organelle connectomic studies, therefore of high interest to cell biologists. The analysis pipeline is carefully designed and described in detail. The results are beautifully visualized and statistically sound. Despite the technical soundness, I am worried about weak links in interpreting the structures in functional context, which I explained in detail in the "Major Points". I recommend the publication of the manuscript after the authors strengthens the structure-function relationships.

We thank the reviewer for these constructive comments. We have now added functional connectivity testing which correlates well with the structural data as well as added several citations and text changes to improve the clarity of our statements.

Major Points:

1. It is unclear which of many structural features obtained in the study is critical to the energy distribution. The candidate features include the arrangement of the mitochondrial network or subnetwork and/or intermitochondrial junction (IMJ). Especially, some of these features indicate different electrical connection. For instance, the mitochondrial network in oxidative muscle lies in two directions while the IMJs were highly ordered perpendicular to the contraction axis. Therefore, it is unclear whether the electric connectivity in oxidative muscle is in two directions or focused in perpendicular direction.

To clarify the issue, I suggest the authors to perform fluorescence imaging with membrane potential sensor such as TMRM as in Ref. 20 and Ref. 30. Especially, longitudinal vs. vertical coupling as in Fig. 3 in Ref. 30 will help answering the directionality of electric connection in oxidative muscle.

We have now performed the functional connectivity testing along the perpendicular and parallel muscle fiber axes in isolated oxidative and glycolytic fibers using TMRM and the photoactivatable

uncoupler as in Ref 20 and (now) 31. As now shown in Supplemental Figure 3, the apparent electrical connectivity in oxidative fibers is similar along both axes as demonstrated by the magnitude of depolarization as a function of distance outside the photoactivated region. Additionally, the functional connectivity in glycolytic muscle is lower than in oxidative muscle as demonstrated by the lower magnitude of depolarization outside the photoactivated region. Further, the functional connectivity was greater along the perpendicular axis than along the parallel axis in glycolytic muscle. Thus, functional connectivity appears to match the structural connectivity based on both network orientation and on subnetwork size. Oxidative muscle is equally connected along both axes and is more connected than glycolytic muscle which is more connected along the perpendicular axis. Consistent with these results, we previously showed (ref 31) that functional connectivity in the cardiac muscle was greater along the parallel axis. Additionally, because there was no apparent difference in directional functional connectivity in oxidative muscle, we conclude that IMJ conductivity, which is primarily oriented along the parallel axis, does not appear to limit connectivity, but rather, the size and orientation of the mitochondrial subnetworks is the primary determinant of the capacity to distribute the membrane potential.

2. It is also unclear which of structural features is linked to spreading of dysfunction. The overall network, subnetwork, IMJ and individual mitochondria can contribute in different ways. I suggest the authors to perform the experiments in Fig.5 in Ref. 30.

Just as with any network (electrical, computer, social, etc.), the spreading of dysfunction is intrinsically linked to the size of the connected network. The functional connectivity tests performed in the new Supplemental Figure 3 are also, by nature, a test of the spread of dysfunction. We depolarize the interior of the cell and measure the depolarization (i.e. dysfunction) in the adjacent regions. Thus, the spread of dysfunction appears to be closely linked to the orientation and size of the mitochondrial subnetworks. As shown in ref 31, this dysfunction outside of the photoactivated region is temporary as there appear to be network protection mechanisms in play. One of the proactive mechanisms is to limit the size of the mitochondrial subnetwork which will structurally limit the spread of dysfunction as shown in Supplemental Figure 3. However, as how the network physically changes in response to damage (i.e. reactive protection mechanisms) is not a focus of this work, the function of the reactive protective mechanisms is outside the scope here.

3. In page 6, 8 and 9, the author stated that the “larger and longer” mitochondria reflect a greater ATP production and energy distribution capacity at the individual mitochondrial level. Is there any supporting data or previous works? In my opinion, these features are unlikely to correlate directly with the functions. It is because the size and length of mitochondrial outer membrane is not linearly correlated to the area of inner membrane. Since the inner membrane is where oxidative phosphorylation occurs and the membrane potential is applied, the total area of inner membrane may be better correlated to the energy production and distribution capacity than the overall size of the organelle outline. For instance,

apoptotic mitochondria are large and dilated while the inner membrane is unfolded and the membrane potential is lowered. Thus, the length may be indicative of high membrane potential, but the volume may be oppositely correlated to the electrical state.

I suggest to quantify the total area of inner membrane per mitochondria using their segmented mitochondrial interior in Supp. Fig. 2c. Alternatively, if the authors show that the density of inner membrane is uniform across the samples, it will become reasonable that the interior area is correlated to the size and length of the organelle.

We have now included several citations from the literature showing that oxidative phosphorylation capacity and protein composition is similar across different striated muscle types when normalized to mitochondrial volume in addition to two references demonstrating that there is also no difference in inner membrane (cristae) surface area between muscle types. Thus, it is reasonable to consider mitochondrial volume as a valid measure of the oxidative phosphorylation capacity.

4. “Mitochondria-SR contact site” is of high interest in cell biology due to its multiple important actions of calcium cycling, lipid trafficking and mitochondrial fission. Because calcium ion diffusion is so fast, the contact site is usually defined as <30 nm gap between the two organelles. However, I can’t find the definition of the contact sites in the text and method. Please specify the definition. Also, using the agreeable definition, segment the contact sites as in IMJ and report the patterns and dimensions.

Mitochondrial contacts sites are defined here as within 30 nm of a mitochondrion. Originally defined only in the methods in the Mitochondrial Network Interaction section, we have now added this definition in the main text as well.

There appears to be no specific pattern to the mito-SR contact sites in relation to the mitochondrial network as projection images of the contact sites look similar to mitochondrial outer membrane projection images. In other words, the orientation of the mito-SR contacts sites is similar to that of the mitochondrial network in a given cell. Additionally, we attempted to differentiate between longitudinal (fiber parallel) and junctional (perpendicular) SR-mito contact sites by using the average distance from the Z disks in oxidative muscle to select each region since junctional SR is always closer to the Z-disk near the I-band while the longitudinal SR is always near the middle of a sarcomere near the A band. However, our initial analyses revealed no differences and pursuit of this analysis was not continued.

In the manuscript, “proximity” is used to compare the contact sites in the three muscles. However, the average proximities of hundreds of nanometers in page 8 are unlikely correlated to the functional contacts because only <30 nm proximity is functional.

Proximity is meant to provide a measure of diffusion distances and not direct, functional contact. While we focused more on direct contact sites, diffusion of metabolites and ions also play a fundamental role in many biological processes. Thus, the distance between different cellular structures is also important for understanding how cells function. We have now clarified that proximity is meant as a measure of diffusion distance in the text.

5. The manuscript ends with the suggestion that specialized mitochondria (i.e. longer, larger mitochondria contacting with lipid droplet have greater capacity for energy conversion and distribution while non-lipid droplet connected mitochondria has greater capacity for calcium cycling) as mechanism for achieving tailored functions of mitochondrial network. To support the statement, there should be supports from previous publications, or functional measurements of membrane potential and dysfunction as in my points #1 and 2, or more relevant analysis of existing data set as in my points #3 and 4.

In addition to the new functional connectivity measurements and citations describing similar oxidative phosphorylation capacity per mitochondrial volume in different muscle types as discussed above, we have now added reference to recent work (Cell Metabolism 2018) from the Shirihi lab at UCLA where they were able to isolated lipid contacting and non-contacting mitochondria separately from brown adipose tissue. They found that lipid droplet associated mitochondria have different bioenergetics, composition, and mitochondrial dynamics than non-lipid droplet associated mitochondria.

Minor Comments:

1. Ref. 20 also claims “automated segmentation”. Compare the current method to the previous ones explicitly, rather than simply referring to the old one as “manual segmentation”.

We thank the reviewer for this comment. Our “automated” segmentation method in the original 2015 work was simply intensity thresholding and some spatial filtering which was sufficient to provide a qualitative overview of what mitochondria look like in the cell. However, there were far too many errors to do any quantitative analysis on the “automated” segmentations resulting in many hours spent tracing mitochondrial structures. We have now added a sentence clarifying this difference.

2. Page 4. 1st paragraph, last sentence. “... representing a time savings of more than 250-fold”. 250-fold comparing to what?

By doing the training on a single XY, XZ, and YZ image plane, we do the training on 0.4% of the pixels (for a 500x1000x1000 pixel image: 500×1000 (XY) + 500×1000 (XZ) + 1000×1000 (YZ) = 2×10^6 out of 5×10^8 total pixels). Since we do not trace all the pixels in the training region (often less than half), we are tracing less than 0.4% of the pixels to segment 100% of the pixels. 100 divided by 0.4 is 250.

3. Page 5. Add definition of “subnetwork”.

We defined a subnetwork as two or more adjacent mitochondria and have added this to the text here.

4. Page 5. For the quantified number of mitochondria per subnetwork, specify the number of subnetwork or total mitochondria to show the sample size.

We have now included the total number of subnetworks and datasets assessed in the Figure 1 legend.

5. Page 7. Last sentence. “the few donut holes found in oxidative and cardiac mitochondria were filled completely by lipid droplets”. What is the total number of donut holes counted? “Completely” is not statistically meaningful without the same size.

In the updated manuscript, there are 196 total donuts counted, 144 were in the glycolytic muscles, 26 in oxidative muscle, and 26 in cardiac muscle. These numbers are now included in the Figure 4 legend. We have changed the text in this section to remove the word completely. In the glycolytic muscle, the sarcoplasmic reticulum and cytosol appear to go through the donut holes. In oxidative and cardiac muscle, lipid droplets appear fill the entire hole such that no other structure can go through.

6. Page 9. “... consistent with the calcium cycling demands of the respective muscles”. Give references.

References are now included with this statement.

7. Page 9. “mitochondria-lipid droplet contacts”. What is the definition used in this work?

We used the same definition of a contact site for both lipid droplets and sarcoplasmic reticulum. If the organelle is within 30 nm of the mitochondria, it is defined as a contact site as stated in the Mitochondrial Network Interactions section of the methods and now also in the main text.

8. Page 10. “... genetically encode tags for electron microscopy in order to assess individual protein interactions with cellular structures at this scale.”

The EM tags involve with radicals of small molecules for increasing local electron density. Since the diffusion speed of small molecule is quite high, the radical spreads out tens of nanometers, which is far larger than molecular scale. Also, each tag produces multiple radicals, so distinguishing individual protein is near impossible. Immunogold EM or super-resolution fluorescence microscopy may be better suited for addressing individual protein interactions.

We agree that achieving individual protein resolution is near impossible and have clarified our statement as that was not our intent. Using the genetic EM tags may still be useful in visualizing

cellular structures which are regularly not visible (e.g. microtubules in our data) or structures which are visible but not clearly distinguishable from other nearby structures.

Reviewer #2 (Remarks to the Author):

In this manuscript, authors present an automated segmentation approach to separate all visually discernable components within the cell. They validate their approach in three muscle cell types and show the difference in mitochondrial network and connectivity in these different cell types.

The manuscript presents significant methodological advance that will be clearly valuable for 3D EM. The limitation of the manuscript is that it is limited to development of new methodological approach and do not present scientific advance beyond that.

We thank the reviewer for these comments. 3D EM/connectomics is essentially another big data field such as proteomics, genomics, metabolomics, etc., but the tools to access the data have not been well developed and much of the data within each dataset is currently left behind. While the major contribution here is the new methodological approach, we do also provide new information on the interactions of mitochondrial donuts with sarcoplasmic reticulum and lipid droplets as well as the presence of specialized lipid connected mitochondria which appear to have different morphological characteristics than non-lipid connected mitochondria. These new results are in line with recent findings from the Shirihai lab (Benador et al, Cell Metabolism, 2018) which also show functional differences between lipid associated and cytosolic mitochondria in brown fat cells and suggest that a better understanding of lipid-connected mitochondria may be critical for the development of new therapies for lipotoxic tissues such as during type II diabetes.

Comments:

Tukey's post hoc test and t-test assume a Gaussian distribution. Authors do not mention whether their data was tested for normal distribution.

We have now included a Brown-Forsythe test to assess the equality of group variances for all analyses comparing the three muscle types by one-way ANOVA. If no difference in variance was found, a Tukey's post hoc test was used. If variances were different, a Dunn's multiple comparisons test was performed. For the lipid connected versus not connected analyses, a Shapiro-Wilks normality test was used. A t-test was performed for datasets which passed the normality test and a Wilcoxon signed rank test was used for datasets which did not pass the normality test.

N is very low for cardiac samples (n=2 as mentioned in Figure 1 and 4) and it seems that also that all other measurements are made only from 2 cardiac samples. Low sample number does not allow to take into account the sample to sample variability.

We have now added analysis of another cardiac dataset to better account for sample to sample variability.

Numerical data is presented only in bar charts. Authors should show the data points using either scatter charts or supplemental excel files.

All data is now shown as scatter plots of the means of each dataset to show the reader all the data that the statistical analyses were performed on.

Reviewer #3 (Remarks to the Author):

Bleck and coauthors in “Connectomics within a Single Cell: Energy Networks in Striated Muscle” present a morphological description and comparison of mitochondrial “networks” in three different striated muscle types and an improved version of technology they previously used to study mitochondrial structure with electron microscopy. Altogether, I do not recommend this paper to be published in Nature Communications because of methodical shortcomings, data overinterpretation and marginal novelty.

We thank the reviewer for this careful review. We hope that the addition of functional connectivity experiments as well as greater clarification regarding data interpretation, statistics, and novelty have assuaged the major concerns listed below.

#1

A critical problem with the presentation and interpretation of data is that the authors interpret any mitochondrial outer membrane apposition in between adjacent mitochondria as electrically conductive, syncytium forming connection. There is no published or currently presented data supporting this. The existence of intermitochondrial junctions (IMJs) is known for about 4 decades, but their functional role, and whether or not they are able to mediate, or are involved in forming electric networks of mitochondria remains unknown. This includes previous work of the authors (Glancy et al., 2015, 2017) that shows a limited spread of photo-induced depolarization of mitochondria indicating a level of limited mitochondrion to mitochondrion communication. Importantly, this cannot prove the involvement of IMJs, because apposed and lumenally continuous mitochondria are not discernible in optical microscopy, plus the authors did not control for uncaged DNP diffusion or localized damage by the uncaging chemistry in these earlier works that may provide an alternative explanation for the observations. Thus electric conduction through IMJs remains a speculation, as the authors noted in their

earlier papers:

“However, the precise physical nature as well as the frequency and conductivity of these specialized IMJs remain unclear.” In Glancy et al 2017 and “We speculate that the EDCS are also conductive elements specifically within the PVM and FPM pools.” In Glancy et al 2015. Speaking of the currently reviewed paper, therefore the functional interpretation of structural findings is merely speculative.

We have now performed directional testing of mitochondrial membrane potential connectivity of oxidative and glycolytic fibers (new Supplemental Figure 3). These results show that the magnitude and direction of functional connectivity is consistent with the orientation and the size of the mitochondrial subnetwork structures. Oxidative fibers are connected similarly along both axes and glycolytic fibers, while less connected than oxidative fibers, are more connected along the perpendicular axis. Similarly, we previously showed (ref 31) that cardiac muscle mitochondria are more functionally connected along the parallel axis similar to the structural connectivity shown here. Electrical connectivity is indeed limited because the mitochondrial network is comprised of many subnetworks which are not physically connected to one another. Due to our segmentation approach, we can now quantify the sizes of the mitochondrial subnetworks in each muscle type. We have now shown that the size and direction of functional connectivity (Supplemental Figure 3) is consistent with the size and orientation of the mitochondrial subnetworks (Figure 1).

We cannot directly test that IMJs are conductive as there is no known way to isolate them for functional testing and the potential IMJ channel proteins are not known though the search is ongoing. Additionally, as the reviewer points out, discriminating between adjacent and lumenally continuous mitochondria using mitochondrial matrix probes such as TMRM in live cells is not clear due to the resolution limits of optical microscopy. To circumvent this, we attempted to use STED super-resolution microscopy of muscle fibers with an outer mitochondrial membrane fluorophore (MitoQC mice- McWilliams et al, JCB, 2016). Unfortunately, even with super resolution, we were not able to clearly discern individual mitochondria due to the axial (z) resolution still likely being larger than the size of the small tubular, 3D mitochondrial shapes in skeletal muscle fibers. However, we routinely see mitochondrial depolarization at least 10 μm away from the photoactivated region in oxidative and cardiac muscles (as in new Supplemental Figure 3 and previously in refs 20 and 31) suggesting mitochondrial functional connectivity persists at least 10 μm within these cells. Of the 5849 individual, lumenally connected mitochondria assessed in the current study, only 6.1% are longer than 5 μm . Thus, lumenally connected mitochondria alone cannot explain the lengths of the functional connectivity measured here and previously. Again, without the current segmentation routine, this type of high-throughput analysis of individual mitochondrial lengths could not have been performed previously. As the only other physical structure that could permit connectivity over the distances measured, it is logical to suggest that the IMJs are involved. Also, as previously, and as pointed out by the reviewer, we remain careful to not state directly that all IMJs are conductive without the direct evidence. However, conductivity of the IMJs appears to be the only logical conclusion in light of the given evidence. Thus, while structure does not directly equal function, measuring IMJ physical size, orientation, and frequency may provide a measure of capacity for function, just as measuring the

physical size and orientation of mitochondrial subnetworks provides a measure of capacity that is consistent with the measured functional connectivity (as shown in Supplemental Figure 3).

We have done extensive controls for the photoactivatable depolarization experiments as reported previously in refs 20 and 31. The shared depolarization outside the irradiation region, as shown here in Supplemental Figure 3 and previously, cannot be due to DNP diffusion. As shown in Figure 4 and associated supplemental movies in ref 31, there is a rapid recovery of the membrane potential outside the irradiated region within about 10 seconds. If DNP was released from the irradiated region and then diffused to and bound to mitochondria in the adjacent region, these mitochondria would stay depolarized rather than quickly recover. We have also controlled for photodamage by 1) performing controls without the photoactivatable agent which show only a small 5-10% photobleach and no spreading outside the irradiated region (Extended Data Figure 8 in ref 20 and Figure 3E in ref 31), 2) purposefully causing photodamage using eight times the light dose used for photoactivation (Movie S3 in ref 31) which caused no immediate shared depolarization of the adjacent regions, and 3) using 405 nm light, which does not uncage the depolarization agent, at the same time as 355 nm light, which does uncage the depolarization agent, in photoswitchable cardiomyocytes (Figure S3 in ref 31). Both the 405 nm and 355 nm light cause photoswitching of the MitoDendra fluorophore in these experiments, but only the 355 nm light causes depolarization and the eventual physical remodeling due to the uncoupling. This shows that even in the presence of the photoactivatable agent, the light intensity used does not cause any detrimental effects on the cell, but rather, it is specific to mitochondrial depolarization. Thus, we are confident that the depolarization events in live cells now reported here and previously are due to functional mitochondrial connectivity and not due to some reaction due to the optical nature of the experiment.

#2

The novelty of the manuscript is marginal. The methodology is not novel, FIB/SEM and SBF/SEM has been used before to visualize mitochondria (including the authors), and also automated segmentation of mitochondria (Márquez Neila et al., 2016). The authors did not develop but applied and optimized a publicly available technology. The comparison of three different muscle tissue goes beyond previous published data from the authors in terms of detail level of description of mitochondrial network and morphology, but in general not a top tier novelty.

We apologize for omitting the citation of Marquez Neila et al. who did report on automated segmentation of mitochondria previously as did Perez et al (as cited in original manuscript), and we have now added this citation. However, Marquez Neila et al., just as Perez et al., are only able to determine whether a pixel within an image is a mitochondrion or not. Their methods are not able to explicitly segment all the mitochondria as individual objects, and as such, in striated muscle or any other cell type where many mitochondria may be tightly packed together, the previous segmentation routines would simply group all adjacent mitochondria together as one. As a result, the previous methods preclude large scale analysis of individual mitochondrial structures as well as intermitochondrial structural connectivity (Figures 2-3). We are also not aware of any previous reports

which evaluate the interactions and connectivity between several organelles within a single cell at this scale (Figure 4). As biologists are increasingly investigating cells as integrated systems rather than individual components, understanding the physical interactions and proximity of different cellular components is critical to our understanding of these complex systems.

#3

Statistics is unclear and possibly wrong. For what are the n-values? Some n-values are very high (100-1000) suggesting that mean \pm SE values were calculated per mitochondrion basis. It is clearly incorrect to use populations of individual mitochondria to draw conclusions on comparing different preparations. Systematic errors, e.g. fixation, sample handling and imaging artifacts will affect the entire population. N-values and respective standard errors must reflect independent experiments. Putting n=100 mitochondria in a statistical comparison will result in a false confidence corresponding to 100 experimental replicates (e.g. using 100 mice)!

We thank the reviewer for this comment. We have now changed all bar graphs to scatter plots to show the means of each dataset while including the total number of mitochondria or other structures analyzed in the figure legends where appropriate. Also, statistical analyses are now performed only on the mean values for each dataset instead of on all the individual mitochondria.

#4

Some comparisons were made using different imaging techniques. E.g. in Fig 1 glycolytic muscle are from SBF-SEM and other from FIB-SEM. Can this bias the comparison of differing tissues?

We included an SBF-SEM dataset to demonstrate the feasibility of our segmentation and analysis routine with another commonly used 3D EM technique which does not have the same resolution in the x, y, and z-axis (6x6x35nm). The mitochondrial network in this glycolytic muscle dataset showed the characteristic perpendicular orientation with respect to the contraction axis as well as similar morphological characteristics of the individual mitochondria and intermitochondrial junctions. While the lower resolution in the z dimension of the SBF-SEM dataset did not prohibit analysis of mitochondrial structures, smaller cellular structures (<100 nm in diameter) such as the sarcoplasmic reticulum were not able to be segmented accurately throughout the entire volume. Thus, the SBF-SEM dataset was only used for mitochondrial specific analyses but not for the mitochondrial interactions with other cellular structures.

#5

The conditions of fixation were very different between the heart and the skeletal muscle. This potentially confounds any conclusion on comparison of these tissues. E.g. Ca²⁺ was chelated with EGTA in the heart before perfusion fixation, while skeletal muscle was fixed in whole. Could observed donut shaped mitochondria (w/o lipid droplet) be due to fixation artifacts?

Fixation conditions were chosen to fix the muscle as close to the non-contracted, in vivo state as possible. The observed non-lipid droplet donut shapes are not likely to be due to fixation because both the oxidative and glycolytic muscles were fixed in the same way and only the glycolytic muscles have them in abundance. Moreover, one of the FIB-SEM datasets contains both an oxidative and glycolytic fiber side by side. In this case of identical fixation and imaging conditions, the glycolytic fiber has five times as many donuts as the oxidative fiber (30 vs 6).

#6

Imprecise or lacking definitions

- Pg 3, line 8 from bottom: “connectomics approach” (and “connectomics framework” on pg 9 middle): please explain what it means. SFig 1 describes mostly image analysis, and “Network Connectivity and Components Analysis” remains unexplained.

Connectomics is a field of research which examines high resolution connectivity between biological structures over large areas. Similar to other -omics fields (proteomics, transcriptomics, metabolomics, etc.), it is characterized by large amounts of data which necessitates the development of high-throughput analytical tools in order to evaluate and assess the data. To date, the developed connectomics analytical tools involve segmentation of cellular structures from 3D image volumes and have primarily been used to evaluate the connectivity of neuronal cells. Here, we adapt the general connectomics approach (image segmentation and connectivity analysis) for use in single cells to evaluate mitochondrial structural connectivity.

Network Connectivity and Components Analysis is the analysis of mitochondrial networks, individual mitochondria within the networks, the junctions connecting mitochondrial networks, and mitochondrial interactions with other cellular structures and is described in the corresponding Figure S1 legend. The specific details of each analysis are included in the methods section.

- Pg 5, ln 10-14: mitochondrial networks, physical connection and intermitochondrial junctions need to be more precisely defined. These definitions are murky and confusingly used through the manuscript. It should be clear where do authors indicate luminal continuity vs. apposition of mitochondrial outer membranes without luminal continuity. The current study is purely structural, so no inferences can be made to electrical connectivity. Based on previous works from the authors and others, apposition of mito membranes is not equal to IMJ or electrical connectivity. I suggest to create a distinct term for the observed “mitochondrial networks” that comprise of packed, but not lumenally continuous mitochondria, that reflects the uncertainty of electrical connectivity.

We have now clarified the text that the mitochondrial network in each muscle type is comprised of many subnetworks that are not physically connected to each other. A subnetwork is two or more (up to hundreds) of individual mitochondrial which are adjacent (i.e. there are no pixels in between) to each other. The physical size of a subnetwork does reflect the maximum capacity for electrical

connectivity but does not demonstrate electrical connectivity as stated by the reviewer. Additionally, we have now shown in Supplemental Figure 3 that mitochondrial subnetwork size and orientation are reflective of the functional connectivity within muscle mitochondrial networks. Combined with Figure 3 from ref 31, we have now shown that mitochondrial electrical connectivity is consistent with mitochondrial subnetwork size and direction in each of the muscle types for which structures are assessed here.

#7

Speculative overinterpretation of the data is frequent through the paper. Observed morphological differences of mitochondria (defined by the outer membrane), without the knowledge of inner membrane structure or protein expression are associated with a wide range of function:

We have now included several citations from the literature showing that oxidative phosphorylation capacity and protein composition is similar across different striated muscle types when normalized to mitochondrial volume in addition to two references demonstrating that there is also no difference in inner membrane surface area per volume between muscle types. Thus, it is reasonable to consider mitochondrial volume as a valid measure of the oxidative phosphorylation capacity in the striated muscles evaluated here.

- Abstract, “Recently, electrical connectivity within muscle mitochondrial networks was demonstrated to provide a rapid mechanism for cellular energy distribution.” – this is an overstatement of the previous results of the authors. Based on those data, some level of electrical connectivity can be concluded, but cannot be stated in a generalized manner.

As in our previously peer reviewed paper titled, “Mitochondrial reticulum for cellular energy distribution in muscle”, we have again demonstrated electrical connectivity within mitochondrial networks in striated muscle cells. In combination with our new segmentation approach, we are now able to show that electrical connectivity is consistent with the mitochondrial subnetwork size of the different muscle types in both in magnitude and direction.

- Pg. 6. Ln 7 from bottom (and pg 9 first paragraph): “suggesting a greater capacity” transporters specific to metabolites and ions are in the mitochondrial inner membrane. To suggest a greater capacity for interaction, the knowledge of surface of the inner membrane, density of cristae junctions and the surface density of relevant carrier proteins would be required. The authors overinterpret the data.

We have now expanded the text in this section to include more references and discussion on the supporting literature behind the suggested greater transport capacity in glycolytic muscle, particularly related to calcium transport. As discussed two points above, inner membrane surface area has been shown to be similar between different striated muscle types. Also, we previously found no differences between oxidative and glycolytic muscle mitochondria in the protein content for 4 (of 4

detected) cristae junction (MICOS) proteins, 13 (of 13 detected) TIM/TOM transport proteins, and in the mitochondrial calcium uniporter (MCU) by mass spectrometry (Glancy et al. AJP Cell, 2011). Thus, a greater surface area to volume ratio with a similar inner membrane surface area and MCU content would logically suggest (but not demonstrate) a greater capacity of glycolytic muscle for interacting with the surrounding environment with respect to calcium. This suggestion is supported by the much larger sarco(endo)plasmic reticulum ATPase (SERCA) content, the greater calcium concentrations during muscle contraction, and the greater ability to uptake calcium in glycolytic muscle (Picard et al. AJP Cell, 2012, Carroll et al. J Physiol 1997, and Fieni et al. Nat Comm, 2012). Further, voltage dependent anion channel (VDAC), one of the most abundant mitochondrial proteins in striated muscle, is an outer membrane transporter which plays a major role in the regulation of metabolic flux as well as cell death which would suggest that outer membrane surface area (as directly measured here) is also an important morphological parameter in addition to the inner membrane surface.

- Pg 7 In 10-11 from below: “priority”, “higher dynamic capacity” what do these mean, and how are they concluded from data? This statement sounds like an overinterpretation of the author’s previous data. **We have removed the use of both “priority” and “higher dynamic capacity” from this section.**

Minor points:

The “null” floating point pixel value may require some explanation for lay reader (to not to confuse with zero).

At the first use of “null” in the text, we have clarified that null values are ignored from all further mathematical analyses.

Pg 17 In 13 Please clarify the filter used. Likely you used some kind of maximum or morphological dilation filter, but not an averaging filter. I don’t expect an averaging filter resulting in the described changes in pixel values.

We used a 1x1x1 pixel 3D mean filter as stated. Because all the pixels within an individual mitochondrion in our analysis have the same value, a 1 pixel mean filter will have no effect on the values of the pixels in the interior (e.g. the mean of 5, 5, 5, 5 = 5). Thus, only the 1 pixel width edges of the mitochondria may be affected by the filter. Mitochondrial edge pixel values which are not adjacent to another mitochondrion are also not altered by this filter because all non-mitochondrial pixels were set to null values (i.e. they are ignored in all mathematical analyses). Therefore, the only pixel values within a mitochondrion that will change are those that are directly adjacent to another mitochondrion which will have different corresponding pixel values.

Pg 18 top: IMJs are surfaces. How were orientation angles defined? In a projection?

Yes, IMJ orientation relative to the muscle contraction axis was determined from projection images. As stated in the original *Intermitochondrial Junctions Analysis* section of the methods, “IMJ

orientation was determined similarly to mitochondrial network orientation described above. The binarized IMJ image was converted to 8-bits and rotated to align the muscle contraction axis horizontally. Then a maximum projection image was created using the Z Project tool and the OrientationJ Distributions plugin was used to determine IMJ angles. IMJs oriented $\pm 0\text{-}29^\circ$ were considered parallel, $\pm 30\text{-}59^\circ$ diagonal, and $\pm 60\text{-}90^\circ$ perpendicular." Representative IMJ maximum projection images were/are shown in Figure 3d-f. While overall orientation (Figure 3g) was assessed in projection images, we have also assessed individual IMJ areas (Figure 3h) and IMJ area per mitochondrial surface area (Figure 3j).

References

- Glancy, B. et al. (2015) 'Mitochondrial reticulum for cellular energy distribution in muscle.', *Nature*, 523(7562), pp. 617–20. doi: 10.1038/nature14614.
- Glancy, B. et al. (2017) 'Power Grid Protection of the Muscle Mitochondrial Reticulum.', *Cell reports*. ElsevierCompany., 19(3), pp. 487–496. doi: 10.1016/j.celrep.2017.03.063.
- Márquez Neila, P. et al. (2016) 'A Fast Method for the Segmentation of Synaptic Junctions and Mitochondria in Serial Electron Microscopic Images of the Brain', *Neuroinformatics*, 14(2), pp. 235–250. doi: 10.1007/s12021-015-9288-z.

Reviewers' comments:

Reviewer #1 (Remarks to the Author):

The revised manuscript has addressed my concerns, particularly by supplementing experimental and literature evidences for linking structural and functional connectivity. The new Supplementary Figure 3 proves that the direction and size of mitochondrial subnetwork in the three different muscle types are consistent to the direction and magnitude of electrical connectivity and spread of dysfunction. Additional citations support the author's interpretation of mitochondrial size and lipid-droplet contacts to different mitochondrial functions. Although I am satisfied with the revision in general, I have a couple suggestions on the text and a couple minor points. Once these points are addressed, I recommend the publication of the second revised version in Nature Communications.

1. I suggest adding a discussion paragraph to summarize which kind of structural features associates to which functional aspects. Although the manuscript discusses functional context whenever it introduces a new structural feature (subnetwork, individual mitochondrial features, IMJ, inter-organellar contacts etc), these discussions are scattered throughout the manuscript and thus inefficient in persuading the potential users of the technique's power. A dedicated discussion paragraph right before the last paragraph of the main text would help readers to envision what this new technique can do for their own research.

2. Both subnetwork and intermitochondrial junction (IMJ) are obtained from adjacent (i.e. "no pixel between neighboring mitochondria") mitochondria. However, their functional roles appear to be quite different. Subnetwork size appears to correlate to electrical connectivity while IMJ seem to relate to chemical signaling. The functional difference may stem from the difference of the definitions of the two structural aspects, but it is unclear in the current manuscript. In the discussion paragraph that I suggested in my point #1, I would like to see discussions explaining this functional difference.

Followings are minor points.

1. Figure 4(k-o) caption. State which symbol (square and triangle) denotes which condition. Also, it is not stated what is the role of lines connecting squares and triangle (my guess is the same set of images).

2. When defining subnetwork as two or more "adjacent" mitochondria, clearly state the dimension of the "adjacency" (i.e. pixel size). This is to make it clear that mitochondrial connections are defined more stringently than the contacts between mitochondria and other organelles.

Reviewer #2 (Remarks to the Author):

The authors have adequately addressed my concerns.

Reviewer #3 (Remarks to the Author):

The authors have improved the paper by sufficiently addressing most, but not all comments of the three referees. I remain concerned of the following items. Besides the technological innovative merit, generic audience will read and cite this paper as a biology paper demonstrating and comparing mitochondrial connectivity in the studied muscle cell types, therefore it is essential that assertions and conclusions regarding mitochondrial structure and function are sound. While the paper is excellent in its structural approach and findings, comes short in supporting claims on function by data. The revised manuscript remained pretentious and speculative regarding

structure-function relationships.

1)

Abstract: "We demonstrate that mitochondrial network orientation, individual mitochondrial size and shape, and the junctions connecting mitochondria within each network are all tuned to meet the differing demands of contraction in each muscle type."

Differing (sub)network geometry has been demonstrated, but not how these are tuned, and how well these meet the demand of contraction. The author's statement is therefore mere speculation.

2)

Page 5: "Each of these mitochondrial networks was comprised of many adjacent mitochondria physically connected through specialized intermitochondrial junctions"

Here I reiterate my original #1 point, because my concerns remain. While it has been addressed in the rebuttal letter, and it is also defined in the methods, please address this explicitly in the main text. Please give in text explicitly that IMJs are defined as apposing mitochondrial outer membranes, and not functionally.

3)

New data has been added to the manuscript to support functional connectivity, but I would be cautious with its interpretation, at least at the current level of analysis. I am generally concerned that the author's anticipation on DNP diffusion are incorrect. Although the author's data is consistent with the well-controlled original observations of the Hartley lab, who developed mitoPhotoDNP (and the author's former peer-reviewed papers), that there is a permanent mitochondrial depolarization at the location of uncaging, it has been never addressed how a permanent depolarization is possibly mediated by uncoupling by DNP, a molecule that diffuses fast across membranes. Physicochemical properties of DNP dictate that it diffuses across membranes, this is the way it uncouples, by shuttling protons between the two sides of the mitochondrial inner membrane. It is well demonstrable that uncoupling by (plain) DNP is reversible (it can be washed out of entire cell cultures). Therefore a transient DNP effect in the areas surrounding the illuminated zone, is highly possible. So the question arises, what is a more likely explanation for the observations; a transient mitochondrial depolarization by DNP diffusion out of the irradiation zone, or the presence of volatile IMJs, that have the ability to form aqueous, electrically conducting pores, unprecedentedly spanning four membranes, and have no molecular identity. Therefore the DNP diffusion as possibility needs to be carefully excluded.

The new SFig 3. may fulfill this role, if data analysis is completed. The authors compared the directionality of DNP uncaging-evoked mitochondrial depolarization in glycolytic and oxidative muscle fibers. The author's conclusion, that the directionality differs is based on that the traces corresponding to glycolytic muscle (blue solid and dotted lines) are statistically different from each other. This needs to be supported by statistical testing. A further concern with these data are that traces do not converge to 1 at larger distances, but to lower values suggesting that the entire cell's mitochondrial population depolarizes (or the plasma membrane potential does) in some extent upon irradiation. This is a particular concern with the comparison of directionality in the glycolytic fibers, where the two traces (if statistically significantly) look like shifted vertically in their entire length. Thus there is a possibility, that not the directionality, but the effect of uncaging on the whole cell was different between the parallel and perpendicular runs. Was the irradiated surface area identical between the two directions? Furthermore, the decrease in mito/cyto ratio should be the same at 0 distance for the perpendicular and parallel recordings to make them comparable, or data need to be normalized to this point. In summary, the newly added data, at least on the current level of analysis is insufficient to support that author's conclusions.

Alternatively, to avoid misinterpretation of a DNP diffusion artifact, images should be taken at a later time, not immediately after uncaging, as given in the methods. As far as I understand, the reason of not doing this is IMJs are thought to subsequently break down and isolate the damaged sub-network. The observed drop in Mito/Cyto Ratio of TMRM in the irradiated area is to 0.6-0.8, this is equivalent to 6-14 mV depolarization (Nernst eq.) that unlikely to be of a pathological magnitude. By decreasing irradiation, if IMJs exist, a lower exposure/uncaging level could be found where the mitochondrial depolarization outside of the exposed area persists as IMJs persist.

4)

There is an internal contradiction overarching the manuscript. Conclusions such as "mitochondrial

structures determined here likely reflect functional capacity" are based on the assumption that mitochondrial function is constant per volume, while the final suggestion (end of abstract, end of main text) suggest specialized roles and functional heterogeneity. If the latter is the conclusion, doesn't it invalidate the analysis? The literature now included (28,29,34-36) to justify this aspiration for a structure-function relationship, describes a functional capacity to volume relationship on tissue level that mitochondrial capacities correlate with the subcellular volume occupied by the mitochondria, and proteomic differences are minimal when normalized to mitochondrial volume. However, these findings should not be extrapolated, and restated for subcellular homogeneity of mitochondria. A counter example (although in a different tissue, brown adipose) is the now cited recent paper from the Shirihai lab showing significant functional and composition differences between mitochondria in differing subcellular locations. Thus, that cardiac mitochondria are larger, is an interesting observation, but data presented here do not tell what kind of functional consequences this has.

Does "mitochondrial structures determined here likely reflect functional capacity." mean Functional units?

5)

Pg 7. "greatest surface area to volume ratio suggesting a greater capacity for these mitochondria to interact with the surrounding environment". As the authors detailed it in their rebuttal letter, this is suggested by an assumed major role of VDAC in controlling mitochondrial metabolite fluxes. Please indicate this assumption in text with appropriate citation. It is important because the other alternative is that carriers in the mitochondrial inner membrane exert major control over metabolism, and cited literature suggests that mitochondrial inner membrane area is proportional to volume, thus not to the outer membrane surface to volume ratio.

6)

Page 5 bottom. The authors find increasing subnetwork size as the mitochondrial volume fraction of the cell type increases. Are IMJs a simple corollary of lots of mitochondria packed in a tight space, so mitochondria are pushed against each other forming outer membrane appositions? Is there a way to decide, e.g. by modeling, whether these membrane appositions occur by chance due to higher volume fractions, or are they organized in any specific manner? This would be a useful addition to the work.

The importance of the above is that the authors suggest the following structure-function relationship on Page 6 "...large, connected networks offer advantages for communication and distribution, but also have the high potential for rapidly spreading dysfunction. Indeed, glycolytic fibers, which are most likely to be damaged during exercise, have the smallest mitochondrial subnetworks". It is just as possible that the glycolytic fibers, which have the lowest mitochondrial volume fractions have the smallest mitochondrial networks, simply for geometric reasons, unrelated to the propensity to damage.

6)

Page 10: "indicates a lower capacity for calcium cycling" Why not "protein and lipid trafficking, and other critical cell processes" as mentioned on the previous page? No data presented to assert what it indicates. Furthermore, being engaged in an activity or not, does not define the capacity to perform the activity.

7)

Page 10: "larger volumes available to convert the stored fatty acids to ATP and are longer and in greater contact with adjacent mitochondria (Figure 4m) suggesting a greater capacity to distribute energy throughout the mitochondrial network." – If larger mitochondria are advantageous for energy distribution, what is the role of IMJ that the authors suggest do not have limiting conductance (rebuttal letter pg 2)? If IMJs are prevalent and conduct proton motive force well, then it will be indifferent how big individual mitochondria are. This is an internal inconsistency.

Minor points:

Pg 6 "While the overall mitochondrial network structures likely reflect the balance between the dynamic needs for force production and to maintain energy homeostasis in each muscle type, the size of the mitochondrial subnetworks may also reflect the propensity of each muscle type to face

damage or dysfunction, as large, connected networks offer advantages for communication and distribution, but also have the high potential for rapidly spreading dysfunction." Balance between what? Needs for force production and energy homeostasis? Or between these former two and the propensity for damage? Please rephrase.

Definitions of scale bars are missing through the supplementary material

Response to Reviewers

Reviewer #1 (Remarks to the Author):

The revised manuscript has addressed my concerns, particularly by supplementing experimental and literature evidences for linking structural and functional connectivity. The new Supplementary Figure 3 proves that the direction and size of mitochondrial subnetwork in the three different muscle types are consistent to the direction and magnitude of electrical connectivity and spread of dysfunction. Additional citations support the author's interpretation of mitochondrial size and lipid-droplet contacts to different mitochondrial functions. Although I am satisfied with the revision in general, I have a couple suggestions on the text and a couple minor points. Once these points are addressed, I recommend the publication of the second revised version in Nature Communications.

1. I suggest adding a discussion paragraph to summarize which kind of structural features associates to which functional aspects. Although the manuscript discusses functional context whenever it introduces a new structural feature (subnetwork, individual mitochondrial features, IMJ, inter-organellar contacts etc), these discussions are scattered throughout the manuscript and thus inefficient in persuading the potential users of the technique's power. A dedicated discussion paragraph right before the last paragraph of the main text would help readers to envision what this new technique can do for their own research.

We thank the reviewer for this suggestion and agree that a dedicated summary of the structure-function relationships provides clarity on the power of this technique. New text has been added prior to the last paragraph of the main text as suggested.

2. Both subnetwork and intermitochondrial junction (IMJ) are obtained from adjacent (i.e. "no pixel between neighboring mitochondria") mitochondria. However, their functional roles appear to be quite different. Subnetwork size appears to correlate to electrical connectivity while IMJ seem to relate to chemical signaling. The functional difference may stem from the difference of the definitions of the two structural aspects, but it is unclear in the current manuscript. In the discussion paragraph that I suggested in my point #1, I would like to see discussions explaining this functional difference.

Discussion of the differences between subnetworks and IMJs has now been added to the new structure-function paragraphs.

Followings are minor points.

1. Figure 4(k-o) caption. State which symbol (square and triangle) denotes which condition. Also, it is not stated what is the role of lines connecting squares and triangle (my guess is the same set of images).

We have now added to the figure legend that squares indicate lipid connected and triangles indicate not connected mitochondria. As the reviewer suggests, the connecting lines indicate paired values from within the same dataset and this information is now included in the figure legend.

2. When defining subnetwork as two or more “adjacent” mitochondria, clearly state the dimension of the “adjacency” (i.e. pixel size). This is to make it clear that mitochondrial connections are defined more stringently than the contacts between mitochondria and other organelles.

We have now indicated at the introduction of the IMJ analysis in the text that they were determined as adjacent pixels, or within 10 nm.

Reviewer #2 (Remarks to the Author):

The authors have adequately addressed my concerns.

Reviewer #3 (Remarks to the Author):

The authors have improved the paper by sufficiently addressing most, but not all comments of the three referees. I remain concerned of the following items. Besides the technological innovative merit, generic audience will read and cite this paper as a biology paper demonstrating and comparing mitochondrial connectivity in the studied muscle cell types, therefore it is essential that assertions and conclusions regarding mitochondrial structure and function are sound. While the paper is excellent in its structural approach and findings, comes short in supporting claims on function by data. The revised manuscript remained pretentious and speculative regarding structure-function relationships.

We thank the reviewer for this careful review. We hope additional clarity on the structure-function relationships guided by the reviewer’s comments has improved this work. Additionally, we now include several references below demonstrating that the diffusion of molecules within striated muscles occurs more readily down the fiber parallel axis. While this is consistent with the cardiac mitochondrial network orientation, it is not consistent with the oxidative muscle mitochondrial network, and it is in opposition to the orientation of the glycolytic muscle mitochondrial network. Thus, DNP diffusion does not seem to be a plausible explanation as to why the orientation of functional mitochondrial connectivity matches that of mitochondrial structural connectivity in three different network types.

1)

Abstract: “We demonstrate that mitochondrial network orientation, individual mitochondrial size and shape, and the junctions connecting mitochondria within each network are all tuned to meet the differing demands of contraction in each muscle type.”

Differing (sub)network geometry has been demonstrated, but not how these are tuned, and how well these meet the demand of contraction. The author’s statement is therefore mere speculation.

We have now changed the phrase “tuned to meet” to “consistent with” the contractile demands which we believe is supported by our data. The heart beats constantly at relatively low power outputs. Glycolytic muscle contracts with high power for short duration. Oxidative muscle falls in the intermediate range on both power and duration. These contractile demands are supported in large part by both energy metabolism and calcium cycling. The constant nature of the beating heart requires constant, high levels of ATP production which is largely met by the mitochondria. The larger mitochondria, more connected mitochondria, and larger mitochondrial subnetworks in the heart are all suggested to provide a greater energy distribution capacity and, thus, are consistent with the contractile demand. Conversely, the short, high power bouts of glycolytic muscle are supported in large part by glycolysis rather than mitochondria and result in the highest calcium cycling demands (both frequency and concentration). The small mitochondria with few connections to adjacent mitochondria making up small subnetworks are consistent with the glycolytic nature of these muscles. The more abundant SR-mito contacts and the greater surface area to volume ratio of the glycolytic muscle is consistent with the calcium cycling demands of this muscle type.

2)

Page 5: “Each of these mitochondrial networks was comprised of many adjacent mitochondria physically connected through specialized intermitochondrial junctions”

Here I reiterate my original #1 point, because my concerns remain. While it has been addressed in the rebuttal letter, and it is also defined in the methods, please address this explicitly in the main text.

Please give in text explicitly that IMJs are defined as apposing mitochondrial outer membranes, and not functionally.

We have now stated in the text here that the intermitochondrial junctions are between opposing outer mitochondrial membranes as suggested.

3)

New data has been added to the manuscript to support functional connectivity, but I would be cautious with its interpretation, at least at the current level of analysis. I am generally concerned that the author’s anticipation on DNP diffusion are incorrect. Although the author’s data is consistent with the well-controlled original observations of the Hartley lab, who developed mitoPhotoDNP (and the author’s former peer-reviewed papers), that there is a permanent mitochondrial depolarization at the location of uncaging, it has been never addressed how a permanent depolarization is possibly mediated by uncoupling by DNP, a molecule that diffuses fast across membranes. Physicochemical properties of DNP dictate that it diffuses across membranes, this is the way it uncouples, by shuttling protons between the two sides of the mitochondrial inner membrane. It is well demonstrable that uncoupling by (plain) DNP is reversible (it can be washed out of entire cell cultures). Therefore a transient DNP effect in the areas surrounding the illuminated zone, is highly possible. So the question arises, what is a more likely explanation for the observations; a transient mitochondrial depolarization by DNP diffusion out of the irradiation zone, or the presence of volatile IMJs, that have the ability to form aqueous, electrically conducting pores, unprecedentedly spanning four membranes, and have no molecular identity. Therefore the DNP diffusion as possibility needs to be carefully excluded.

While we cannot directly exclude the possibility of rapid DNP diffusion during these experiments because DNP is not fluorescently tagged, it is very unlikely to explain the results shown in Supplemental Figure 3. Both cardiac and skeletal muscle are known to have anisotropic diffusion characteristics with diffusion occurring more easily along the fiber parallel axis. This has been shown for oxygen (Homer et al. AJP Regu, 1984), calcium (Engel et al. Biophys J, 1994), ATP (Vendelin et al. AJP Cell, 2008), and even photons (Binzoni et al. Phys. Med. Biol, 2006) among others. This greater propensity for diffusion along the fiber parallel axis was thus a concern we discussed in Glancy et al. Cell Reports, 2017 when looking at functional connectivity of the fiber parallel oriented cardiac mitochondrial network. However, in the functional connectivity experiments here, the connectivity is either similar along both axes (oxidative muscle) or greater along the fiber perpendicular axis (glycolytic muscle). Thus, the directionality of functional coupling in glycolytic fibers is in the wrong direction to be explained by diffusion of DNP.

The new SFig 3. may fulfill this role, if data analysis is completed. The authors compared the directionality of DNP uncaging-evoked mitochondrial depolarization in glycolytic and oxidative muscle fibers. The author's conclusion, that the directionality differs is based on that the traces corresponding to glycolytic muscle (blue solid and dotted lines) are statistically different from each other. This needs to be supported by statistical testing.

We have now compared the parallel and perpendicular depolarization schemes in the glycolytic muscle at 0, 1, 3, 5, 7, and 9 μm away from the irradiated region by two-tailed, two sample, equal variance t-test. The mitochondria are significantly more depolarized in the perpendicular coupling scheme at 3, 5, 7, and 9 μm suggesting functional connectivity at greater distances than along the parallel axis. These differences are now indicated on the graph in the supplemental figure.

A further concern with these data are that traces do not converge to 1 at larger distances, but to lower values suggesting that the entire cell's mitochondrial population depolarizes (or the plasma membrane potential does) in some extent upon irradiation. This is a particular concern with the comparison of directionality in the glycolytic fibers, where the two traces (if statistically significantly) look like shifted vertically in their entire length. Thus there is a possibility, that not the directionality, but the effect of uncaging on the whole cell was different between the parallel and perpendicular runs. Was the irradiated surface area identical between the two directions? Furthermore, the decrease in mito/cyto ratio should be the same at 0 distance for the perpendicular and parallel recordings to make them comparable, or data need to be normalized to this point. In summary, the newly added data, at least on the current level of analysis is insufficient to support that author's conclusions.

The lack of convergence to 1.0 is likely because of slight photobleaching due to the repeated imaging scheme as even control experiments with no DNP present do not converge to 1.0 away from the irradiated region (see Figure 3F from Glancy et al. Cell Reports, 2017). The irradiated region of interest was not statistically different between the perpendicular and parallel glycolytic experiments ($530 \pm 6 \mu\text{m}^2$ vs $515 \pm 18 \mu\text{m}^2$, respectively, $p = .371$ by two-tailed, two sample, equal variance t-test). Thus, it is not likely that there were differential effects of uncaging on the whole cell between these two experiments. The magnitude of the decrease in the mito/cyto ratio at distance zero is indicative of

the degree of connectivity (a disconnected network would see no decrease outside the irradiated region), thus, normalizing to this value would result in the loss of this valuable information.

Alternatively, to avoid misinterpretation of a DNP diffusion artifact, images should be taken at a later time, not immediately after uncaging, as given in the methods. As far as I understand, the reason of not doing this is IMJs are thought to subsequently break down and isolate the damaged sub-network. The observed drop in Mito/Cyto Ratio of TMRM in the irradiated area is to 0.6-0.8, this is equivalent to 6-14 mV depolarization (Nernst eq.) that unlikely to be of a pathological magnitude. By decreasing irradiation, if IMJs exist, a lower exposure/uncaging level could be found where the mitochondrial depolarization outside of the exposed area persists as IMJs persist.

Post/pre TMRM ratio images 55 and 75 seconds after the irradiation are shown in Figure 4 in Glancy et al. Cell Reports 2017 in heart and skeletal muscle fibers showing that after the initial shared depolarization between the irradiated and adjacent regions, the irradiated region continues to depolarize fully while the adjacent regions quickly (5-10 seconds) recover their electrical potential. As suggested by the reviewer, we also showed in that paper that the mitochondrial network within the irradiated region eventually breaks down and physically separates from the rest of the network in a manner inconsistent with mitochondrial fission. We agree that finding a method to induce a state where the shared mitochondrial depolarization persists is of great interest to the study of IMJs and their role in protection of the mitochondrial network. However, we believe detailed analyses of potential IMJ functions are worthy of a separate study and is outside the scope of this largely structural work.

4)

There is an internal contradiction overarching the manuscript. Conclusions such as “mitochondrial structures determined here likely reflect functional capacity” are based on the assumption that mitochondrial function is constant per volume, while the final suggestion (end of abstract, end of main text) suggest specialized roles and functional heterogeneity. If the latter is the conclusion, doesn't it invalidate the analysis? The literature now included (28,29,34-36) to justify this aspiration for a structure-function relationship, describes a functional capacity to volume relationship on tissue level that mitochondrial capacities correlate with the subcellular volume occupied by the mitochondria, and proteomic differences are minimal when normalized to mitochondrial volume. However, these findings should not be extrapolated, and restated for subcellular homogeneity of mitochondria. A counter example (although in a different tissue, brown adipose) is the now cited recent paper from the Shirihai lab showing significant functional and composition differences between mitochondria in differing subcellular locations. Thus, that cardiac mitochondria are larger, is an interesting observation, but data presented here do not tell what kind of functional consequences this has.

Does “mitochondrial structures determined here likely reflect functional capacity.” mean Functional units?

The literature cited to support the structure-function relationships also includes two papers showing that mitochondrial cristae density is similar per mitochondrial volume in different muscle fiber types

based on electron microscopy which provides greater than tissue level resolution. Moreover, one of these papers (Hoppeler et al. *J Physiol* 1987) shows that mitochondrial inner membrane surface area is the same across different subcellular regions and across different fiber types. Thus, subcellular homogeneity per mitochondrial volume has already been demonstrated in muscle cells. Regarding the lipid connected versus not connected mitochondrial analysis, we do not suggest that there are differences between the types when normalized to mitochondrial volume. We show that the lipid droplet-connected mitochondria have more volume than non-connected mitochondria. Thus, based on the data from Hoppeler at the subcellular level and others at the single cell or tissue level showing similar oxidative phosphorylation capacity per mitochondrial volume, we conclude that the lipid droplet-connected mitochondria have a greater oxidative phosphorylation capacity simply because they have more volume.

5)

Pg 7. “greatest surface area to volume ratio suggesting a greater capacity for these mitochondria to interact with the surrounding environment”. As the authors detailed it in their rebuttal letter, this is suggested by an assumed major role of VDAC in controlling mitochondrial metabolite fluxes. Please indicate this assumption in text with appropriate citation. It is important because the other alternative is that carriers in the mitochondrial inner membrane exert major control over metabolism, and cited literature suggests that mitochondrial inner membrane area is proportional to volume, thus not to the outer membrane surface to volume ratio.

We do not dispute that the mitochondrial inner membrane exerts control over metabolism. In the previous review from this reviewer, regarding the same statement about interaction capacity quoted here, it was suggested that similarity of the inner membrane surface area and the density of the carrier proteins was required for our statement to be valid. In our response, we provided several literature citations in support of this. VDAC was mentioned in the rebuttal simply to state that not all control is in the inner membrane as was implied by the original review, but this does not mean that the inner membrane has no control over metabolism as suggested here. Mitochondrial flux control has been shown to be shared across the entire energy conversion pathway (e.g. Korzeniewski, *Biochem J*, 1996 and Glancy et al. *Biochemistry* 2013). Please see Hodge and Colombini, *J. Membrane Biol.* 157, 271–279 (1997) and Guzun et al. *Biochimica et Biophysica Acta* 1818 (2012) 1545–1554 for two examples discussing the role of VDAC in metabolic flux control.

6)

Page 5 bottom. The authors find increasing subnetwork size as the mitochondrial volume fraction of the cell type increases. Are IMJs a simple corollary of lots of mitochondria packed in a tight space, so mitochondria are pushed against each other forming outer membrane appositions? Is there a way to decide, e.g. by modeling, whether these membrane appositions occur by chance due to higher volume fractions, or are they organized in any specific manner? This would be a useful addition to the work. The importance of the above is that the authors suggest the following structure-function relationship on Page 6 “...large, connected networks offer advantages for communication and distribution, but also have

the high potential for rapidly spreading dysfunction. Indeed, glycolytic fibers, which are most likely to be damaged during exercise, have the smallest mitochondrial subnetworks". It is just as possible that the glycolytic fibers, which have the lowest mitochondrial volume fractions have the smallest mitochondrial networks, simply for geometric reasons, unrelated to the propensity to damage.

The IMJs are not simply due to volume packing of mitochondria as the increased electron density characteristic of IMJs is not apparent in EM images of mouse liver and kidney (Xu et al. Biochem J, 2008) which have mitochondrial volumes much higher than in the mouse skeletal muscles used here. This was also discussed previously in Glancy et al. Cell Reports, 2017.

We agree that it is possible that the smaller mitochondrial networks in glycolytic fibers could simply be due to geometric reasons and have removed the statement about spreading dysfunction.

6)

Page 10: "indicates a lower capacity for calcium cycling" Why not "protein and lipid trafficking, and other critical cell processes" as mentioned on the previous page? No data presented to assert what it indicates. Furthermore, being engaged in an activity or not, does not define the capacity to perform the activity.

We have now changed "indicates" to "suggests." We chose to focus on calcium cycling instead of the many other functions ascribed to mito-SR contacts due to its importance for muscle contraction.

7)

Page 10: "larger volumes available to convert the stored fatty acids to ATP and are longer and in greater contact with adjacent mitochondria (Figure 4m) suggesting a greater capacity to distribute energy throughout the mitochondrial network." – If larger mitochondria are advantageous for energy distribution, what is the role of IMJ that the authors suggest do not have limiting conductance (rebuttal letter pg 2)? If IMJs are prevalent and conduct proton motive force well, then it will be indifferent how big individual mitochondria are. This is an internal inconsistency.

We thank the reviewer for this comment. One potential role of IMJs is to provide dynamic regulation of connectivity within a network that must face varying cellular conditions. If the network was one big mitochondrion, damage at any point could take down the entire network. By having many mitochondria connected by IMJs, a dysfunctional region of the network can be isolated within 5-10 seconds (Glancy et al. Cell Reports, 2017). In that case, why not have a network made of many very small, highly connected mitochondria? One potential reason is that smaller structures of the same shape will have a greater surface area to volume ratio (e.g. S/V of a sphere equals 3 divided by the radius). For a mitochondrion, that means the outer membrane would make up a greater proportion of the overall volume. Conversely, for larger mitochondria, the outer membrane would make up a smaller proportion of the overall volume. Indeed, chronic muscle stimulation causing a two-fold increase in mitochondrial content resulted in a lower proportion of mitochondrial outer membrane (but not inner membrane) per unit mitochondrial volume for both subsarcolemmal and intermyofibrillar mitochondria (Reichmann et al. Pflugers Arch, 1985) that was attributed to changes

in mitochondrial size and shape. Thus, changing mitochondrial size results in changes to the surface area to volume of the outer mitochondrial membrane which has implications for cellular interactions as discussed in the manuscript, but also for mitophagy, the process of removal of damaged mitochondria which involves wrapping membrane structures around the damaged mitochondria. This process would require more membranes for mitochondria with relatively greater surface areas. There may be other factors in play, but this is just one example of why individual mitochondrial size is important even if IMJs do not limit conductivity. This discussion has now been added to the text in the new structure-function summary paragraphs requested by Reviewer #1.

Minor points:

Pg 6 “While the overall mitochondrial network structures likely reflect the balance between the dynamic needs for force production and to maintain energy homeostasis in each muscle type, the size of the mitochondrial subnetworks may also reflect the propensity of each muscle type to face damage or dysfunction, as large, connected networks offer advantages for communication and distribution, but also have the high potential for rapidly spreading dysfunction.” Balance between what? Needs for force production and energy homeostasis? Or between these former two and the propensity for damage? Please rephrase.

We have now removed this statement in response to point number six (the first one) above.

Definitions of scale bars are missing through the supplementary material

Definitions of scale bars have now been added to the supplemental figure legends.

Methods: TMRM nm -> nM

Thank you for catching this mistake. It has now been corrected.

REVIEWERS' COMMENTS:

Reviewer #1 (Remarks to the Author):

The re-revised manuscript has properly addressed my concerns.

Reviewer #3 (Remarks to the Author):

The authors have addressed all of my concerns. However, the revised manuscript now states on page 11: "magnitude and direction of functional connectivity correlates strongly with the size and orientation of the mitochondrial subnetworks (Supplemental Figure 3)". This is a new exaggeration of the results. There was no any kind of correlation analysis performed to state this. At best "is consistent with", but not "correlates strongly". Regarding the "strength" of this "correlation", judging based on the error bars and noise in SFig 3m blue dotted line, the arbitrary four points indicated by "*" used for statistical analysis could be arbitrarily taken in another time points and would probably show no statistical difference.

Response to Reviewers

REVIEWERS' COMMENTS:

Reviewer #1 (Remarks to the Author):

The re-revised manuscript has properly addressed my concerns.

Reviewer #3 (Remarks to the Author):

The authors have addressed all of my concerns. However, the revised manuscript now states on page 11: "magnitude and direction of functional connectivity correlates strongly with the size and orientation of the mitochondrial subnetworks (Supplemental Figure 3)". This is a new exaggeration of the results. There was no any kind of correlation analysis performed to state this. At best "is consistent with", but not "correlates strongly". Regarding the "strength" of this "correlation", judging based on the error bars and noise in SFig 3m blue dotted line, the arbitrary four points indicated by "*" used for statistical analysis could be arbitrarily taken in another time points and would probably show no statistical difference.

We have now changed "correlates strongly" to "consistent with" as suggested by the reviewer.